# DBLoss: Decomposition-based Loss Function
# for Time Series Forecasting

**Xiangfei Qiu[1], Xingjian Wu[1], Hanyin Cheng[1], Xvyuan Liu[1]**
**Chenjuan Guo[1], Jilin Hu[1,2][*], Bin Yang[1]**
[1]East China Normal University, [2]KLATASDS-MOE
{xfqiu, xjwu, hycheng, xyliu}@stu.ecnu.edu.cn,
{cjguo, jlhu, byang}@dase.ecnu.edu.cn

## Abstract

Time series forecasting holds significant value in various domains such as economics, traffic, energy, and AIOps, as accurate predictions facilitate informed decision-making. However, the existing Mean Squared Error (MSE) loss function sometimes fails to accurately capture the seasonality or trend within the forecasting horizon, even when decomposition modules are used in the forward propagation to model the trend and seasonality separately. To address these challenges, we propose a simple yet effective **D**ecomposition-**B**ased **Loss** function called **DBLoss**. This method uses exponential moving averages to decompose the time series into seasonal and trend components within the forecasting horizon, and then calculates the loss for each of these components separately, followed by weighting them. As a general loss function, DBLoss can be combined with any deep learning forecasting model. Extensive experiments demonstrate that DBLoss significantly improves the performance of state-of-the-art models across diverse real-world datasets and provides a new perspective on the design of time series loss functions.

**Resources:** https://github.com/decisionintelligence/DBLoss.

## 1 Introduction

Time Series Forecasting holds significant value in various domains such as economics [Wang et al., 2025a, Li et al., 2025a, Ma et al., 2025a, Liu et al., 2025a], traffic [Guo et al., 2014, Wu et al., 2023a, Lu et al., 2011, Yue et al., 2025a, Ma et al., 2025b,c, Liu et al., 2025b], energy [Wang et al., 2025b, Huang et al., 2025a, Ma et al., 2025d, Miao et al., 2024], and AIOps [Wang et al., 2025c, Yue et al., 2024, Ma et al., 2025e, Wu et al., 2024a], as accurate predictions facilitate astute decision-making. To pursue accurate predictions, recent progress in Long-term Time Series Forecasting focuses on effectively capturing the inherent seasonality and trend, which reflect the changing laws of the time series, i.e., the inductive bias. Recently, dozens of deep learning models have been designed from light-weight to multi-scale, such as DLinear [Zeng et al., 2023], OLinear [Yue et al., 2025b], CycleNet [Lin et al., 2024a], TimesNet [Wu et al., 2023b], TimeBase [Huang et al., 2025b], PDF [Dai et al., 2024], TimeMixer [Wang et al., 2024], and DUET [Qiu et al., 2025a], which are aiming at capturing such inductive bias consistently within data for more accurate predictions.

Technically, to capture the seasonality and trend within data, decomposition-based techniques are widely applied to disentangle the seasonality and trend parts explicitly. For example, DLinear and DUET apply the moving-average technique to obtain the trend part. TimesNet, PDF and TimeMixer apply meticulously-designed seasonal decomposition modules to process the seasonal part, and the CycleNet uses a learnable matrix to directly capture the seasonality. All these techniques are

---

[*]Corresponding Author

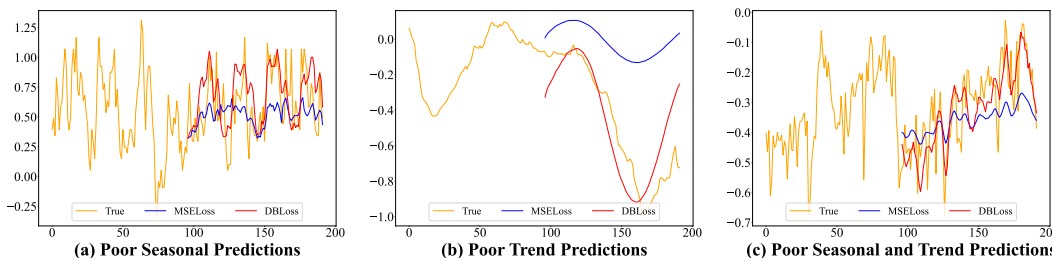

Figure 1: Limitations of MSE loss in capturing seasonality or trend within the forecasting horizon.

employed in the forward propagation to effectively extract the seasonal and trend components from the contextual time series.

However, if the purpose of extracting seasonality and trend in the contextual time series is to improve predictions, perhaps considering seasonality and trends directly in the forecasting horizon may further enhance prediction performance. As shown in Figure 1, we observe that current distance-based loss functions (such as MSE) have the following limitations: 1) they may make poor seasonal predictions; 2) they may make poor trend predictions; 3) they may make both poor seasonal and trend predictions. Even when decomposition techniques are applied in the forward propagation, the seasonality and trend within the forecasting horizon are not effectively modeled, indicating that the inductive bias is not well applied to the predictions.

Inspired by the above motivation, we manage to explicitly encourage the modeling of the seasonality and trend in the forecasting horizon to enhance the performance. Specifically, we propose a simple yet effective **D**ecomposition-**B**ased **Loss** function called **DBLoss**. This method involves using exponential moving averages [Stitsyuk and Choi, 2025] to decompose the time series into seasonal and trend components within the forecasting horizon. It then calculates the loss for each of these components separately and combines them with appropriate weighting. As a general loss function, combining DBLoss with any deep learning forecasting model can lead to consistent improvement in performance, which is demonstrated on real-world datasets from multiple domains. The contributions are summarized as follows.

- We propose a simple yet effective loss function for time series forecasting, called DBLoss, which can refine the characterization and representation of time series through decomposition within the forecasting horizon.

- The proposed DBLoss is generally applicable to arbitrary deep neural networks with negligible cost. By introducing DBLoss into the baseline, we have achieved performance that generally surpasses the state-of-the-art on eight real-world datasets.

- We conduct extensive evaluations of DBLoss using quantitative analysis and qualitative visualizations to verify its effectiveness.

## 2 Related works

### 2.1 Time Series Forecasting Methods

Time series forecasting (TSF) predicts future observations based on historical observations. TSF methods are mainly categorized into four distinct approaches: (1) statistical learning-based methods, (2) machine learning-based methods, (3) deep learning-based methods, and (4) foundation methods.

Early TSF methods primarily rely on statistical learning approaches such as ARIMA [Box and Pierce, 1970], ETS [Hyndman et al., 2008], and VAR [Godahewa et al., 2021]. With advancements in machine learning, methods like XGBoost [Chen and Guestrin, 2016], Random Forests [Breiman, 2001], and LightGBM [Ke et al., 2017] gain popularity for handling nonlinear patterns. However, these methods still require manual feature engineering and model design [Ma et al., 2025f, Wang et al., 2023, Wu et al., 2025a]. Leveraging the representation learning of deep neural networks (DNNs) [Huang et al., 2023, Miao et al., 2025, Wang et al., 2025d], many deep learning-based methods emerge. TimesNet [Wu et al., 2023b] and SegRNN [Lin et al., 2023] model time series as vector sequences,

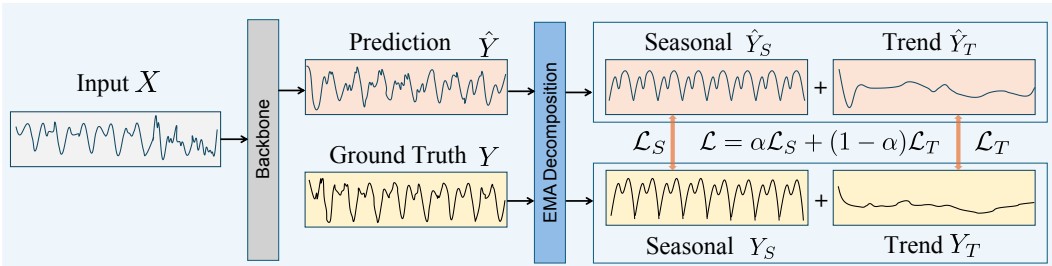

Figure 2: Overview of the proposed DBLoss.

using CNNs or RNNs to capture temporal dependencies. Additionally, Transformer architectures, including DUET [Qiu et al., 2025a], Informer [Zhou et al., 2021], FEDformer [Zhou et al., 2022], Triformer [Cirstea et al., 2022], and PatchTST [Nie et al., 2023], capture complex relationships between time points more accurately, significantly improving forecasting performance. MLP-based methods, including SparseTSF [Lin et al., 2024b], CycleNet [Lin et al., 2024a], SRSNet [Wu et al., 2025b], NLinear [Zeng et al., 2023], and DLinear [Zeng et al., 2023], adopt simpler architectures with fewer parameters but still achieve highly competitive forecasting accuracy.

However, many of these methods struggle with generalization across domains due to their reliance on domain-specific data [Li et al., 2025b]. To address this, foundation methods are proposed, categorized into LLM-based methods and time series pre-trained methods. LLM-based methods [Zhou et al., 2023, Jin et al., 2024, Liu et al., 2024a, Pan et al., 2024] leverage the strong representational capacity and sequential modeling capability of LLMs to capture complex patterns for time series modeling. Time series pre-trained methods [Liu et al., 2024b, Gao et al., 2024, Goswami et al., 2024, Das et al., 2024] focus on pre-training over multi-domain time series data, enabling the method to learn domain-agnostic features that are transferable across various applications. This strategy not only enhances performance on specific tasks but also provides greater flexibility when adapting to new datasets or scenarios.

## 2.2  Loss Functions for Time Series Forecasting

Recently, to enhance the training performance of time series forecasting models, researchers have introduced various novel loss functions. These loss functions can be broadly categorized into three types: shape-based losses, dependency-based losses, and patch-based structural losses.

Shape-based losses aim to capture structural similarities between true values and predictions by tackling the issue of shape mismatch. For example, techniques based on Dynamic Time Warping (DTW), such as Soft-DTW [Cuturi and Blondel, 2017] and DILATE [Le Guen and Thome, 2019], can achieve alignment even when time series undergo deformation. However, despite their excellent performance in improving shape alignment, the high computational complexity of these methods restricts their application in large-scale scenarios. Meanwhile, TILDE-Q [Lee et al., 2022] introduces transformation invariance, making it robust to amplitude shifts, phase changes, and scale differences, thus focusing more on similarity at the shape level. Dependency-based losses are dedicated to characterizing temporal correlations within the forecasting horizon. For instance, FreDF [Wang et al., 2025e] cleverly circumvents complex correlation modeling between labels by performing learning and prediction in the frequency domain. Furthermore, patch-based structural losses like PSLoss [Kudrat et al., 2025] incorporate patch-wise statistical properties into the loss function, enabling a more granular structural measurement of the data. Unlike the aforementioned loss functions, our proposed DBLoss refines the characterization and representation of time series through decomposition within the forecasting horizon, offering a novel perspective for the design of time series loss functions.

## 3  DBLoss

A *time series* $\boldsymbol{X} \in \mathbb{R}^{N \times T}$ is a time-oriented sequence of N-dimensional time points, where $T$ is the number of timestamps, and $N$ is the number of channels. If $N = 1$, a time series is called univariate, and multivariate if $N > 1$. *Time Series Forecasting* aims to predict the next $F$ future timestamps, formulated as $\boldsymbol{Y} = \langle \boldsymbol{X}_{:,T+1}, \cdots, \boldsymbol{X}_{:,T+F} \rangle \in \mathbb{R}^{N \times F}$ based on the historical time

series $\boldsymbol{X} = \langle \boldsymbol{X}_{:,1}, \cdots, \boldsymbol{X}_{:,T} \rangle \in \mathbb{R}^{N \times T}$ with $N$ channels and $T$ timestamps. For convenience, we separate dimensions with commas. Specifically, we denote $\boldsymbol{X}_{i,j} \in \mathbb{R}$ as the $i$-th channel at the $j$-th timestamp, $\boldsymbol{X}_{n,:} \in \mathbb{R}^T$ as the time series of $n$-th channel, where $n = 1, \cdots, N$.

## 3.1 Overview

As shown in Figure 2, we first generate the prediction $\hat{\boldsymbol{Y}}$ using an arbitrary backbone method. Next, we input both the prediction $\hat{\boldsymbol{Y}}$ and the ground truth $\boldsymbol{Y}$ into the EMA Decomposition Module to decompose them into seasonal and trend components. Through this process, we obtain the seasonal component $\hat{\boldsymbol{Y}}_S$ and the trend component $\hat{\boldsymbol{Y}}_T$ of the prediction, as well as the seasonal component $\boldsymbol{Y}_S$ and the trend component $\boldsymbol{Y}_T$ of the ground truth. Subsequently, we compute the errors for both the seasonal and trend components and then combine these errors using a weighted sum to form the final loss function. This approach allows for a more accuracy evaluation of the differences between the predicted and ground truth values, leading to more effective optimization and training.

## 3.2 EMA Decomposition Module

Seasonal-trend decomposition facilitates the learning of complex temporal patterns by breaking down time series signals into trend and seasonal components. Trend components refer to the long-term changes or patterns that occur over time, intuitively representing the overall direction of the data. In contrast, seasonal components capture the phenomena in the time series that repeat at specific intervals and are typically nonlinear due to the complexity and variability of periodic behavior. This technique is widely applied in time series analysis methods [Wu et al., 2021, Zhou et al., 2022, Zeng et al., 2023, Wang et al., 2024, Qiu et al., 2025a]. Unlike above methods, which typically extract the trend and seasonal representations of the time series through decomposition and then combine these two representations to obtain a more comprehensive time series representation for downstream tasks, our DBLoss computes the loss by separately decomposing the prediction and the ground truth into their trend and seasonal components. We then compute the losses for the trend and seasonal components separately and finally combine these losses using a weighted sum. This process enables the model to better capture the trends and seasonality of the ground truth, resulting in more accurate predictions.

There are various methods for seasonal-trend decomposition, such as STL decomposition [Cleveland et al., 1990], Simple Moving Average (SMA) decomposition [Wu et al., 2021, Qiu et al., 2025a, Zeng et al., 2023], and Exponential Moving Average (EMA) decomposition [Stitsyuk and Choi, 2025]. In this study, we chose EMA decomposition. Specifically, after obtaining the prediction $\hat{\boldsymbol{Y}}$ and the ground truth $\boldsymbol{Y}$, we input them into the EMA decomposition module to decompose them into their trend and seasonal components. We then compute the final loss in the weighted loss function described in Section 3.3. Algorithm 1 details the calculation process of EMA decomposition module.

---

**Algorithm 1** Calculation Process of EMA Decomposition Module

---

**Input:** Time series $X \in \mathbb{R}^{B \times T \times N}$, where $B$ is the batch size, $T$ is the time steps, and $N$ is the number of channels; Smoothing factor $\alpha \in (0, 1)$
**Output:** Seasonality and Trend of $X$, denoted as $Seasonality \in \mathbb{R}^{B \times T \times N}, Trend \in \mathbb{R}^{B \times T \times N}$

 1: Get the shape of $X$: $B, T, N \leftarrow X.shape$
 2: Calculate the weights: $W \leftarrow [(1-\alpha)^{T-1}, (1-\alpha)^{T-2}, \cdots, 1]$
 3: Copy the weights to create a divisor: $D_{\text{div}} \leftarrow W.clone()$
 4: Update the weights for EMA calculation: $W[1:] \leftarrow W[1:] \times \alpha$
 5: Reshape the weights and divisor:
 6: $\quad W \leftarrow W.reshape(1, T, 1)$
 7: $\quad D_{\text{div}} \leftarrow D_{\text{div}}.reshape(1, T, 1)$
 8: Compute the cumulative sum of weighted data: $C \leftarrow \text{cumsum}(X \times W, \dim = 1)$
 9: Divide the cumulative sum by the divisor: $Trend \leftarrow \frac{C}{D_{\text{div}}}$
10: Calculate the residual: $Seasonality \leftarrow X - Trend$
11: **return** $Seasonality, Trend$

---

## 3.3 Weighted Loss Function

Based on the EMA decomposition, we obtain the predicted seasonal component $\hat{Y}_S$ and trend component $\hat{Y}_T$, as well as the corresponding ground truth values $Y_S$ and $Y_T$. We then propose a weighted loss function, which consists of two parts: the seasonal loss $\mathcal{L}_S$ and the trend loss $\mathcal{L}_T$.

$$\mathcal{L}_S := \left| \hat{Y}_S - Y_S \right|_2, \ \mathcal{L}_T := \left| \hat{Y}_T - Y_T \right|_1. \tag{1}$$

To prevent the loss of one component from dominating the optimization process due to scale differences, we introduce a **scale alignment** mechanism. Specifically, the trend loss is adaptively adjusted according to the relative magnitude between $\mathcal{L}_S$ and $\mathcal{L}_T$:

$$\mathcal{L}_T^{\text{aligned}} := \mathcal{L}_T \times \text{stopgrad} \left( \frac{\mathcal{L}_S}{\mathcal{L}_T + \epsilon} \right), \tag{2}$$

where $\epsilon$ is a small constant to ensure numerical stability. Here, stopgrad($\cdot$) denotes a **gradient detachment** operation, which prevents the gradient from back-propagating through the alignment ratio, thereby avoiding interference between the two loss components.

Finally, we define the total loss $\mathcal{L}$ as:

$$\mathcal{L} := \beta \cdot \mathcal{L}_S + (1 - \beta) \cdot \mathcal{L}_T^{\text{aligned}}, \tag{3}$$

where $\beta$ is a tuning parameter used to balance the contributions of the seasonal loss and the trend loss. By adjusting the value of $\beta$, we can optimize the model's training process according to specific application scenarios.

We provide a theoretical analysis in Appendix C to explain why the proposed DBLoss is more effective than the conventional MSE loss for time series forecasting.

# 4 Experiments

## 4.1 Setup

**Datasets** To conduct comprehensive and fair comparisons for different models, we conduct experiments on eight well-known forecasting benchmarks as the target datasets, including ETT (ETTh1, ETTh2, ETTm1, ETTm2), Solar, Weather, Electricity, and Traffic. For more details on the benchmark datasets, please refer to Table 5 in Appendix $A$.

**Backbones** We selected eight state-of-the-art (SOTA) time series forecasting models to serve as baselines. Specifically, we include four time series specific models: iTransformer [Liu et al., 2024c], Amplifier [Fei et al., 2025], PatchTST [Nie et al., 2023], and DLinear [Zeng et al., 2023], as well as four time series foundation models: CALF [Liu et al., 2025c], UniTS [Gao et al., 2024], TTM [Ekambaram et al., 2024], and GPT4TS [Zhou et al., 2023].

**Implementation Details** To keep consistent with previous works, we adopt Mean Squared Error (MSE) and Mean Absolute Error (MAE) as evaluation metrics. We consider four forecasting horizon $F$: {96, 192, 336, 720} for all datasets. We utilize the comprehensive time series forecasting benchmark TFB [Qiu et al., 2024] for unified evaluation, with all baseline results also derived from TFB. *Please note that for all the baseline scripts, we directly use the optimal scripts provided by TFB and only replace the training loss function with DBLoss, without making any other modifications.* The purpose of this approach is to validate the effectiveness of DBLoss to the greatest extent possible. By doing so, we can ensure the accuracy of the experimental results and clearly demonstrate the performance improvements brought by DBLoss. We do not apply the "Drop Last" trick [Qiu et al., 2024, 2025b,c] to ensure a fair comparison. All experiments of DBLoss are conducted using PyTorch in Python 3.8 and executed on an NVIDIA Tesla-A800 GPU. The training process is guided by the MSE loss function and employs the ADAM optimizer. The initial batch size is set to 64, with the flexibility to halve it (down to a minimum of 8) in case of an Out-Of-Memory (OOM) issue.

## 4.2 Main results

We present the MSE and MAE of four state-of-the-art long-term multivariate forecasting models on eight real-world datasets in Table 1. Notably, DBLoss observes performance improvements

Table 1: Long-term multivariate forecasting results. The table reports MSE and MAE for different forecasting horizons $F \in \{96, 192, 336, 720\}$. The parameters for the baselines are kept consistent with those of TFB [Qiu et al., 2024]. The better results are highlighted in **bold**.

| Model | | iTransformer | | | | Amplifier | | | | PatchTST | | | | DLinear | | | |
|---|---|---|---|---|---|---|---|---|---|---|---|---|---|---|---|---|---|
| Loss | | Ori | | DBLoss | | Ori | | DBLoss | | Ori | | DBLoss | | Ori | | DBLoss | |
| Metric | | MSE | MAE | MSE | MAE | MSE | MAE | MSE | MAE | MSE | MAE | MSE | MAE | MSE | MAE | MSE | MAE |
| ETTh1 | 96 | 0.386 | 0.405 | **0.383** | **0.396** | **0.376** | 0.393 | **0.376** | **0.389** | 0.377 | 0.397 | **0.373** | **0.390** | 0.379 | 0.403 | **0.369** | **0.390** |
| | 192 | 0.424 | 0.440 | **0.405** | **0.421** | 0.414 | 0.42 | **0.409** | **0.415** | 0.409 | 0.425 | **0.395** | **0.413** | 0.408 | 0.419 | **0.402** | **0.409** |
| | 336 | 0.449 | 0.460 | **0.425** | **0.438** | 0.442 | 0.446 | **0.430** | **0.432** | 0.431 | 0.444 | **0.414** | **0.426** | 0.440 | 0.440 | **0.430** | **0.428** |
| | 720 | 0.495 | 0.487 | **0.478** | **0.463** | 0.48 | 0.479 | **0.459** | **0.465** | 0.457 | 0.477 | **0.425** | **0.451** | 0.471 | 0.493 | **0.449** | **0.475** |
| | Avg | 0.439 | 0.448 | **0.423** | **0.430** | 0.428 | 0.435 | **0.419** | **0.425** | 0.419 | 0.436 | **0.402** | **0.420** | 0.425 | 0.439 | **0.412** | **0.425** |
| ETTh2 | 96 | 0.297 | 0.348 | **0.288** | **0.337** | 0.291 | 0.342 | **0.288** | **0.332** | **0.274** | 0.337 | **0.274** | **0.334** | 0.300 | 0.364 | **0.284** | **0.342** |
| | 192 | 0.372 | 0.403 | **0.357** | **0.389** | 0.355 | 0.4 | **0.344** | **0.379** | 0.348 | 0.384 | **0.334** | **0.376** | 0.387 | 0.423 | **0.357** | **0.390** |
| | 336 | 0.388 | 0.417 | **0.385** | **0.416** | 0.384 | 0.42 | **0.377** | **0.405** | 0.377 | 0.416 | **0.349** | **0.392** | 0.490 | 0.487 | **0.407** | **0.430** |
| | 720 | **0.424** | 0.444 | 0.427 | **0.443** | 0.422 | 0.451 | **0.400** | **0.437** | 0.406 | 0.441 | **0.390** | **0.422** | 0.704 | 0.597 | **0.586** | **0.533** |
| | Avg | 0.370 | 0.403 | **0.364** | **0.396** | 0.363 | 0.403 | **0.352** | **0.388** | 0.351 | 0.395 | **0.337** | **0.381** | 0.470 | 0.468 | **0.409** | **0.424** |
| ETTm1 | 96 | 0.300 | 0.353 | **0.290** | **0.341** | 0.293 | 0.347 | **0.287** | **0.335** | 0.289 | 0.343 | **0.284** | **0.328** | 0.300 | 0.345 | **0.295** | **0.337** |
| | 192 | 0.341 | 0.380 | **0.328** | **0.363** | 0.329 | 0.367 | **0.328** | **0.359** | 0.329 | 0.368 | **0.322** | **0.355** | 0.336 | 0.366 | **0.331** | **0.358** |
| | 336 | 0.374 | 0.396 | **0.368** | **0.386** | 0.365 | 0.387 | **0.364** | **0.380** | 0.362 | 0.390 | **0.359** | **0.376** | 0.367 | 0.386 | **0.361** | **0.378** |
| | 720 | 0.429 | 0.430 | **0.415** | **0.415** | 0.429 | 0.422 | **0.424** | **0.413** | 0.416 | 0.423 | **0.410** | **0.412** | 0.419 | 0.416 | **0.415** | **0.409** |
| | Avg | 0.361 | 0.390 | **0.350** | **0.376** | 0.354 | 0.381 | **0.351** | **0.372** | 0.349 | 0.381 | **0.344** | **0.368** | 0.356 | 0.378 | **0.351** | **0.370** |
| ETTm2 | 96 | 0.175 | 0.266 | **0.166** | **0.254** | 0.168 | 0.258 | **0.163** | **0.245** | 0.165 | 0.255 | **0.163** | **0.246** | 0.164 | 0.255 | **0.163** | **0.247** |
| | 192 | 0.242 | 0.312 | **0.227** | **0.295** | 0.227 | 0.298 | **0.222** | **0.288** | 0.221 | 0.293 | **0.219** | **0.284** | 0.224 | 0.304 | **0.220** | **0.290** |
| | 336 | 0.282 | 0.337 | **0.278** | **0.330** | 0.276 | 0.334 | **0.271** | **0.322** | 0.276 | 0.327 | **0.273** | **0.320** | **0.277** | 0.337 | **0.277** | **0.329** |
| | 720 | **0.375** | 0.394 | **0.375** | **0.388** | 0.364 | 0.394 | **0.350** | **0.373** | 0.362 | 0.381 | **0.357** | **0.374** | 0.371 | 0.401 | **0.366** | **0.390** |
| | Avg | 0.269 | 0.327 | **0.262** | **0.317** | 0.259 | 0.321 | **0.252** | **0.307** | 0.256 | 0.314 | **0.253** | **0.306** | 0.259 | 0.324 | **0.257** | **0.314** |
| Solar | 96 | 0.190 | 0.244 | **0.180** | **0.215** | **0.184** | 0.239 | 0.189 | **0.226** | 0.170 | 0.234 | **0.167** | **0.211** | **0.199** | 0.265 | 0.202 | **0.236** |
| | 192 | **0.193** | 0.257 | 0.201 | **0.239** | **0.202** | 0.252 | 0.208 | **0.239** | 0.204 | 0.302 | **0.182** | **0.226** | **0.220** | 0.282 | 0.224 | **0.250** |
| | 336 | 0.203 | 0.266 | **0.195** | **0.232** | 0.232 | 0.274 | 0.235 | **0.251** | 0.212 | 0.293 | **0.187** | **0.232** | **0.234** | 0.295 | 0.237 | **0.256** |
| | 720 | **0.223** | 0.281 | 0.232 | **0.265** | **0.229** | 0.276 | 0.242 | **0.256** | 0.215 | 0.307 | **0.197** | **0.237** | **0.243** | 0.301 | 0.245 | **0.260** |
| | Avg | **0.202** | 0.262 | **0.202** | **0.238** | 0.212 | 0.260 | 0.219 | **0.243** | 0.200 | 0.284 | **0.183** | **0.227** | **0.224** | 0.286 | 0.227 | **0.251** |
| Weather | 96 | 0.157 | 0.207 | **0.154** | **0.196** | 0.147 | 0.199 | **0.145** | **0.189** | 0.150 | 0.200 | **0.149** | **0.189** | 0.170 | 0.230 | **0.169** | **0.221** |
| | 192 | 0.200 | 0.248 | **0.197** | **0.239** | 0.188 | 0.238 | **0.186** | **0.228** | 0.191 | 0.239 | **0.189** | **0.229** | **0.216** | 0.273 | **0.216** | **0.262** |
| | 336 | 0.252 | 0.287 | **0.249** | **0.278** | 0.239 | 0.276 | **0.239** | **0.269** | 0.242 | 0.279 | **0.240** | **0.270** | 0.258 | 0.307 | **0.253** | **0.293** |
| | 720 | 0.320 | 0.336 | **0.319** | **0.335** | **0.316** | 0.328 | **0.316** | **0.323** | **0.312** | 0.330 | 0.314 | **0.322** | 0.323 | 0.362 | **0.319** | **0.346** |
| | Avg | 0.232 | 0.270 | **0.230** | **0.262** | 0.223 | 0.260 | **0.221** | **0.252** | 0.224 | 0.262 | **0.223** | **0.252** | 0.242 | 0.293 | **0.239** | **0.280** |
| Electricity | 96 | 0.134 | 0.230 | **0.131** | **0.226** | **0.132** | **0.227** | 0.133 | **0.227** | **0.143** | 0.247 | **0.143** | **0.244** | **0.140** | 0.237 | **0.140** | **0.235** |
| | 192 | 0.154 | 0.250 | **0.149** | **0.242** | 0.149 | 0.241 | **0.147** | **0.239** | 0.158 | 0.260 | **0.158** | **0.257** | 0.154 | 0.251 | **0.154** | **0.247** |
| | 336 | 0.169 | 0.265 | **0.163** | **0.257** | 0.165 | 0.258 | **0.163** | **0.256** | 0.168 | 0.267 | **0.165** | **0.259** | 0.169 | 0.268 | **0.169** | **0.264** |
| | 720 | **0.194** | 0.288 | 0.195 | **0.284** | **0.203** | 0.292 | **0.203** | **0.290** | **0.214** | 0.307 | **0.214** | **0.304** | 0.204 | 0.301 | **0.203** | **0.295** |
| | Avg | 0.163 | 0.258 | **0.160** | **0.252** | **0.162** | 0.255 | **0.162** | **0.253** | 0.171 | 0.270 | **0.170** | **0.266** | **0.167** | 0.264 | **0.167** | **0.260** |
| Traffic | 96 | **0.363** | 0.265 | 0.366 | **0.261** | 0.396 | 0.278 | **0.393** | **0.270** | 0.370 | 0.262 | **0.369** | **0.254** | **0.395** | 0.275 | 0.396 | **0.270** |
| | 192 | **0.384** | 0.273 | 0.387 | **0.271** | 0.413 | 0.285 | **0.412** | **0.275** | 0.386 | 0.269 | **0.385** | **0.260** | **0.407** | 0.280 | 0.407 | **0.274** |
| | 336 | **0.396** | 0.277 | 0.397 | **0.275** | 0.421 | 0.291 | 0.422 | **0.286** | 0.396 | 0.275 | **0.395** | **0.266** | 0.417 | 0.286 | **0.415** | **0.279** |
| | 720 | 0.445 | 0.308 | **0.444** | **0.306** | **0.456** | 0.307 | **0.456** | **0.304** | 0.435 | 0.295 | **0.432** | **0.286** | 0.454 | 0.308 | **0.449** | **0.298** |
| | Avg | **0.397** | 0.281 | 0.399 | **0.278** | 0.422 | 0.290 | **0.421** | **0.284** | 0.397 | 0.275 | **0.395** | **0.267** | 0.418 | 0.287 | **0.417** | **0.280** |

across all backbone models and significantly outperforms MSE loss in most cases. This validates the robustness and broad applicability of the proposed loss function. Furthermore, DBLoss achieves significant improvements on models that have already adopted trend-seasonal decomposition to further extract better model representations, such as DLinear [Zeng et al., 2023]. This indicates that performing trend-seasonal decomposition during the loss computation does not conflict with the previous trend-seasonal decomposition operations but rather enhances model performance.

## 4.3 Comparison with Other Loss Functions

To better validate the effectiveness of DBLoss, we compare it with several other loss functions—see Table 2. TILDE-Q emphasizes shape similarity using transformation-invariant loss terms. FreDF cleverly circumvents complex correlation modeling between labels by performing learning and prediction in the frequency domain. PSLoss incorporates patch-wise statistical properties into the loss function, enabling a more granular structural measurement of the data. The results indicate that DBLoss achieves the lowest MSE and MAE in most cases across various datasets and forecasting horizons. This is due to its ability to refine the characterization and representation of time series through decomposition within the forecasting horizon, thereby achieving a more precise alignment between the ground truth and predictions.

Table 2: Comparison between the proposed DBLoss and other loss functions. The model is DLinear and we report the result of three datasets-ETTh2, ETTm1, and Traffic. The best results are highlighted in **bold**, and the second-best results are highlighted in underline.

| Dataset | | ETTh2 | | | | | ETTm1 | | | | | Traffic | | | | |
|---|---|---|---|---|---|---|---|---|---|---|---|---|---|---|---|---|
| Forecast horizon | | 96 | 192 | 336 | 720 | Avg | 96 | 192 | 336 | 720 | Avg | 96 | 192 | 336 | 720 | Avg |
| Ori | MSE | 0.300 | 0.387 | 0.490 | 0.704 | 0.470 | 0.300 | 0.336 | 0.367 | 0.419 | 0.356 | **0.395** | **0.407** | 0.417 | 0.454 | 0.418 |
| | MAE | 0.364 | 0.423 | 0.487 | 0.597 | 0.468 | 0.345 | 0.366 | 0.386 | 0.416 | 0.378 | 0.275 | 0.280 | 0.286 | 0.308 | 0.287 |
| TILDE-Q (2022) | MSE | 0.287 | 0.362 | 0.425 | 0.599 | 0.418 | 0.302 | 0.336 | 0.371 | 0.425 | 0.359 | 0.416 | 0.422 | 0.423 | 0.461 | 0.431 |
| | MAE | 0.345 | 0.395 | 0.445 | 0.551 | 0.434 | 0.342 | 0.362 | 0.386 | 0.417 | 0.377 | 0.294 | 0.296 | 0.293 | 0.316 | 0.300 |
| FreDF (2025e) | MSE | 0.284 | 0.362 | 0.420 | 0.587 | 0.413 | 0.302 | 0.333 | 0.363 | **0.415** | 0.353 | 0.398 | 0.408 | 0.416 | 0.452 | 0.419 |
| | MAE | **0.342** | 0.396 | 0.445 | 0.546 | 0.432 | 0.344 | 0.363 | 0.363 | 0.411 | 0.375 | **0.270** | 0.275 | 0.280 | 0.302 | 0.282 |
| PSLoss (2025) | MSE | **0.283** | 0.358 | 0.411 | 0.614 | 0.417 | 0.296 | 0.332 | 0.361 | 0.416 | 0.351 | 0.398 | 0.408 | 0.416 | 0.452 | 0.419 |
| | MAE | 0.343 | 0.393 | 0.434 | 0.549 | 0.430 | 0.339 | 0.361 | 0.380 | 0.413 | 0.373 | **0.270** | 0.274 | 0.279 | 0.299 | 0.281 |
| DBLoss (Ours) | MSE | 0.284 | 0.357 | 0.407 | 0.586 | 0.409 | 0.295 | 0.331 | 0.361 | 0.415 | 0.351 | 0.396 | 0.407 | 0.415 | 0.449 | 0.417 |
| | MAE | **0.342** | 0.390 | 0.430 | 0.533 | 0.424 | 0.337 | 0.358 | 0.378 | 0.409 | 0.370 | 0.270 | 0.274 | 0.279 | 0.298 | 0.280 |

Table 3: Zero-shot forecasting results on ETT datasets. The forecasting horizon is 720. The parameters for the baselines are kept consistent with those of TFB [Qiu et al., 2024]. The better results are highlighted in **bold**.

| Model | iTransformer | | | | Amplifier | | | | PatchTST | | | | DLinear | | | |
|---|---|---|---|---|---|---|---|---|---|---|---|---|---|---|---|---|
| Loss | Ori | | DBLoss | | Ori | | DBLoss | | Ori | | DBLoss | | Ori | | DBLoss | |
| Metric | MSE | MAE | MSE | MAE | MSE | MAE | MSE | MAE | MSE | MAE | MSE | MAE | MSE | MAE | MSE | MAE |
| ETTh1→ETTh2 | 0.461 | 0.470 | **0.434** | **0.446** | **0.393** | **0.427** | 0.401 | 0.431 | 0.402 | 0.437 | **0.389** | **0.427** | 0.647 | 0.573 | **0.542** | **0.520** |
| ETTh1→ETTm1 | 1.061 | 0.676 | **0.771** | **0.592** | 0.777 | **0.571** | 0.758 | 0.576 | 0.753 | 0.590 | **0.722** | **0.570** | 0.754 | 0.602 | **0.735** | **0.584** |
| ETTh1→ETTm2 | 0.454 | 0.447 | **0.434** | **0.420** | 0.406 | 0.415 | 0.412 | **0.414** | 0.403 | 0.414 | **0.400** | **0.410** | 0.640 | 0.566 | **0.535** | **0.510** |
| ETTh2→ETTh1 | 0.672 | 0.593 | **0.557** | **0.521** | 0.678 | 0.592 | **0.530** | **0.515** | 0.593 | 0.556 | **0.484** | **0.490** | 0.506 | 0.521 | **0.450** | **0.477** |
| ETTh2→ETTm1 | 0.969 | 0.659 | **0.802** | **0.594** | 0.761 | 0.585 | **0.714** | **0.564** | 0.762 | 0.577 | **0.738** | **0.551** | 0.752 | 0.608 | **0.738** | **0.579** |
| ETTh2→ETTm2 | **0.417** | 0.428 | 0.436 | **0.422** | 0.403 | 0.417 | **0.403** | **0.415** | **0.393** | 0.409 | 0.395 | **0.404** | 0.787 | 0.629 | **0.580** | **0.526** |
| ETTm1→ETTh1 | 0.705 | 0.598 | **0.528** | **0.516** | 0.500 | 0.494 | **0.482** | **0.488** | 0.710 | 0.594 | **0.553** | **0.534** | 0.460 | 0.481 | **0.445** | **0.468** |
| ETTm1→ETTh2 | 0.433 | 0.460 | **0.409** | **0.444** | 0.425 | 0.446 | **0.421** | **0.445** | **0.418** | **0.451** | 0.431 | 0.452 | 0.427 | 0.464 | **0.404** | **0.444** |
| ETTm1→ETTm2 | **0.369** | 0.389 | 0.370 | **0.387** | **0.369** | 0.384 | 0.372 | **0.384** | 0.370 | 0.391 | **0.367** | **0.384** | 0.389 | 0.416 | **0.367** | **0.394** |
| ETTm2→ETTh1 | 1.001 | 0.704 | **0.775** | **0.613** | 0.542 | 0.524 | **0.479** | **0.491** | 0.896 | 0.695 | **0.617** | **0.577** | 0.488 | 0.497 | **0.460** | **0.481** |
| ETTm2→ETTh2 | 0.477 | 0.486 | **0.456** | **0.468** | 0.444 | 0.464 | **0.414** | **0.439** | 0.412 | 0.449 | **0.400** | **0.431** | 0.415 | 0.452 | **0.410** | **0.445** |
| ETTm2→ETTm1 | 0.662 | 0.566 | **0.551** | **0.498** | 0.652 | 0.547 | **0.478** | **0.452** | 0.484 | 0.451 | **0.452** | **0.429** | 0.449 | 0.439 | **0.436** | **0.430** |

## 4.4 Zero-shot Forecasting Results

To evaluate the effectiveness of DBLoss in enhancing the generalization ability on unseen datasets, we follow the methods outlined in [Chen et al., 2024, Kudrat et al., 2025] and conduct zero-shot forecasting experiments. Specifically, we sequentially use ETTh1, ETTh2, ETTm1, and ETTm2 as source datasets, while the remaining datasets serve as target datasets.

Table 3 shows the results measured on the target datasets when the forecasting horizon is set to 720. These results highlight the consistent advantages of DBLoss. In most cases, DBLoss outperforms MSE loss, indicating that it can significantly improve the model's generalization performance across different datasets and sampling frequencies. These improvements stem from DBLoss's ability to better capture the intrinsic trends and seasonal patterns within the datasets, thereby enabling the model to more effectively adapt to unseen data patterns.

## 4.5 Results on Time Series Foundation Models

To further evaluate the effectiveness of the proposed DBLoss, we conducted 5% few-shot experiments on four time series foundation models, using DBLoss only during the fine-tuning stage. These models include two LLM-based time series forecasting models: CALF [Liu et al., 2025c], GPT4TS [Zhou et al., 2023], as well as two time series pre-trained models: UniTS [Gao et al., 2024], TTM [Ekambaram et al., 2024]. The results in Table 4 show that incorporating DBLoss consistently outperforms the standard MSE loss. These findings highlight that DBLoss not only enhances performance on specific models but also improves the performance of foundation models, further demonstrating its significant role in multivariate time series forecasting.

Table 4: Foundation models results in the 5% few-shot setting. The table reports average MSE and MAE for four forecasting lengths $F \in \{96, 192, 336, 720\}$. The parameters for the baselines are kept consistent with those of TSFM-Bench [Li et al., 2025c]. The better results are highlighted in **bold**. Full results are provided in Table 10 of Appendix G

| Model | GPT4TS | | | | CALF | | | | TTM | | | | UniTS | | | |
|---|---|---|---|---|---|---|---|---|---|---|---|---|---|---|---|---|
| Loss | Ori | | DBLoss | | Ori | | DBLoss | | Ori | | DBLoss | | Ori | | DBLoss | |
| Metric | MSE | MAE | MSE | MAE | MSE | MAE | MSE | MAE | MSE | MAE | MSE | MAE | MSE | MAE | MSE | MAE |
| ETTh1 | 0.467 | 0.470 | **0.453** | **0.462** | 0.443 | 0.454 | **0.433** | **0.446** | 0.405 | 0.425 | **0.395** | **0.417** | 0.436 | 0.434 | **0.425** | **0.427** |
| ETTh2 | 0.373 | 0.414 | **0.368** | **0.406** | 0.373 | 0.407 | **0.368** | **0.404** | 0.342 | 0.383 | **0.332** | **0.378** | 0.372 | 0.405 | **0.357** | **0.393** |
| ETTm1 | 0.388 | 0.404 | **0.377** | **0.394** | 0.372 | 0.396 | **0.358** | **0.382** | 0.356 | 0.376 | **0.354** | **0.372** | 0.377 | 0.402 | **0.362** | **0.386** |
| ETTm2 | 0.278 | 0.335 | **0.266** | **0.320** | 0.271 | 0.332 | **0.259** | **0.316** | 0.258 | 0.313 | **0.257** | **0.308** | 0.292 | 0.344 | **0.270** | **0.320** |
| Solar | 0.262 | 0.335 | **0.254** | **0.279** | **0.229** | **0.297** | 0.246 | 0.300 | **0.219** | 0.269 | 0.224 | **0.266** | **0.206** | 0.261 | 0.214 | **0.246** |
| Weather | 0.253 | 0.293 | **0.248** | **0.284** | 0.238 | 0.277 | **0.236** | **0.272** | **0.225** | 0.260 | 0.225 | **0.256** | **0.230** | 0.269 | 0.231 | **0.260** |
| Electricity | **0.207** | 0.317 | **0.207** | **0.309** | 0.172 | 0.268 | **0.171** | **0.264** | 0.179 | 0.277 | **0.178** | **0.274** | **0.180** | 0.275 | 0.181 | **0.274** |
| Traffic | 0.433 | 0.309 | **0.428** | **0.295** | 0.435 | 0.316 | **0.433** | **0.309** | 0.484 | 0.341 | **0.481** | **0.339** | 0.422 | 0.289 | **0.420** | **0.282** |

## 4.6 Impact of DBLoss on Generalization

To examine how DBLoss affects training dynamics and generalization capabilities, we use both MSE loss and DBLoss as objective functions and visualize the MSE on the training and testing datasets across all training epochs—see Figure 3. We observe a consistent trend across all datasets. During training, models optimized with only MSE loss have lower errors per epoch compared to those optimized with DBLoss. However, on the test data, models trained with MSE loss exhibit higher errors per epoch than those trained with DBLoss. These observations indicate that while models trained with MSE loss have lower losses during the training phase, they generalize poorly on test data. In contrast, DBLoss enhances the model's generalization and prediction accuracy by encouraging the model to learn the trends and seasonal patterns in the dataset. Additionally, models trained with MSE loss show significant MSE fluctuations in the test loss on some datasets (e.g., Figure 3b and Figure 3d), whereas DBLoss demonstrates greater stability.

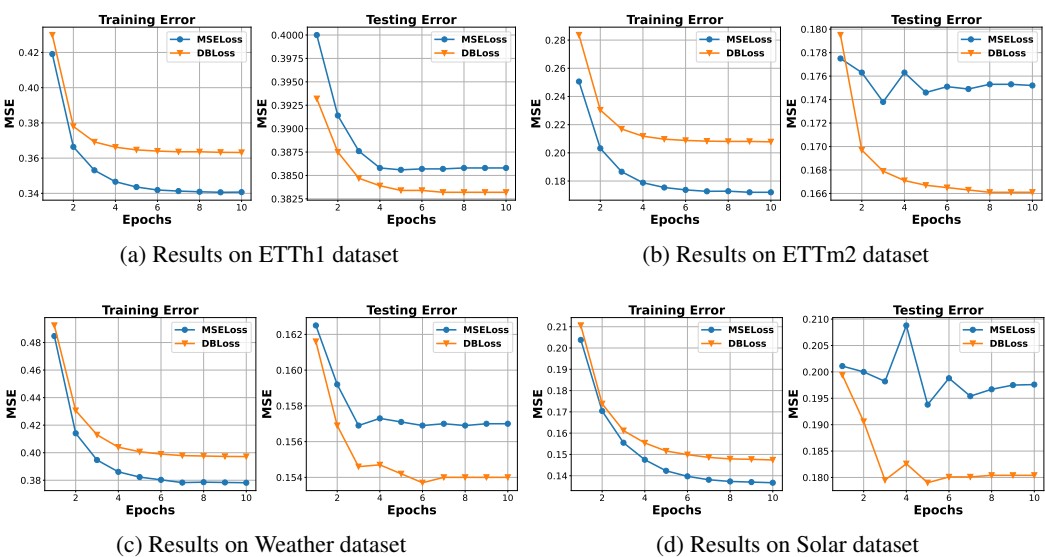

(a) Results on ETTh1 dataset      (b) Results on ETTm2 dataset

(c) Results on Weather dataset      (d) Results on Solar dataset

Figure 3: Training and testing MSE loss curves across all training epochs for the iTransformer model trained with MSE loss and DBLoss on the ETTh1, ETTm2, Weather, and Solar datasets. Notably, the model trained with DBLoss exhibits higher training errors but achieves lower testing errors. This highlights the effectiveness of DBLoss in enhancing generalization and mitigating overfitting.

## 4.7 Hyperparameter Sensitivity

Our method has two hyperparameters: the score weight $\beta$ for weighted loss and the smoothing factor $\alpha$ for EMA decomposition. To handle extreme cases, we manually replace $\alpha = 0$ and $\alpha = 1$ with approximate values close to 0 and 1, respectively. Specifically, a larger $\beta$ increases the proportion of the seasonal component in the loss calculation, while a smaller $\alpha$ results in heavier smoothing, making the trend smoother and the seasonal component more prominent.

Conversely, a larger $\alpha$ results in less smoothing, making the trend less smooth and the seasonal component less noticeable. From Figure 4, we have the following observations: 1) When $\beta$ is too large (e.g., $\beta = 1$) or too small (e.g., $\beta = 0$), the model's performance is poor. 2) For datasets with pronounced seasonality, such as traffic, a larger score weight $\beta$ (i.e., considering a higher proportion of the seasonal component in the loss calculation) yields better performance. A smaller smoothing factor $\alpha$ (i.e., making the seasonal component more

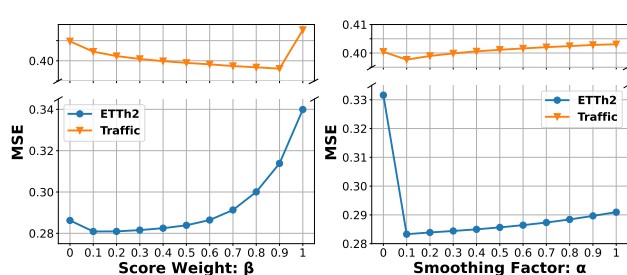

Figure 4: The impact of the hyperparameter on ETTh2 and Traffic based DLinear (horizon 96).

prominent) also improves performance. 3) For datasets with less pronounced seasonality, such as ETTh2, a moderate $\beta$ value (e.g., 0.4 or 0.5) achieves better results, indicating that the proportions of the seasonal and trend components should be balanced. The variation in the smoothing factor $\alpha$ has a minimal impact on performance. 4) However, we find that the optimal values of $\alpha$ and $\beta$ may vary across different algorithms. At present, there is no definitive method for selecting these parameters. We discuss this limitation in Appendix H and leave it as an open problem for future research.

## 4.8 Forecasting Visualization

Figure 5 shows the visualization of forecasting results for samples from the ETTh1 dataset. We can observe that the predictions obtained with DBLoss are closer to the ground truth. This is mainly because DBLoss encourages the model to better learn the seasonal and trend patterns in the dataset. More visualization results are provided in Appendix D.

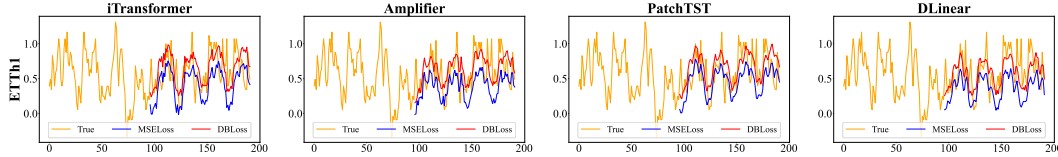

Figure 5: Forecasting visualization comparing DBLoss and MSE loss as objective functions.

## 5 Conclusion

In this study, we propose DBLoss to address the traditional MSE that sometimes fails to accurately capture the seasonality or trend within the forecasting horizon, even when decomposition modules are used in the forward propagation to model the trend and seasonality separately. Specifically, our method uses exponential moving averages to decompose the time series into seasonal and trend components within the forecasting horizon, and then calculates the loss for each of these components separately, followed by weighting them. By introducing DBLoss into the baseline model, we have achieved performance that surpasses the state-of-the-art on eight real-world datasets. Additionally, all datasets and code are available at https://github.com/decisionintelligence/DBLoss.

## Acknowledgments and Disclosure of Funding

This work was partially supported by the National Natural Science Foundation of China (No. 62472174), the Open Research Fund of Key Laboratory of Advanced Theory and Application in Statistics and Data Science–MOE, ECNU, the Fundamental Research Funds for the Central Universities, and the ECNU Multifunctional Platform for Innovation (001).

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

# A Datasets

To conduct comprehensive and fair comparisons for different models, we conduct experiments on eight well-known forecasting benchmarks as the target datasets, including (I) **ETT** (Electricity Transformer Temperature, 4 subsets) data contains seven features of electricity transformer data collected from two separate counties between July 2016 and July 2018. These datasets vary in granularity, with "h" indicating hourly data and "m" indicating 15-minute intervals. The suffixes "1" and "2" refer to two different regions from which the datasets originated. (II) **Weather** data includes 21 meteorological factors recorded every 10 minutes in 2020 at the Max Planck Biogeochemistry Institute's Weather Station. (III) **Electricity** data contains hourly electricity consumption data of 321 clients from 2012 to 2014. (IV) **Solar** data records the solar power production of 137 PV plants in 2006, which are sampled every 10 minutes. (V) **Traffic** data contains road occupancy rates measured by 862 sensors on freeways in the San Francisco Bay Area from 2015 to 2016, recorded hourly.

Table 5: Statistics of datasets.

| Dataset | Domain | Frequency | Lengths | Dim | Split | Description |
|---|---|---|---|---|---|---|
| ETTh1 | Electricity | 1 hour | 14,400 | 7 | 6:2:2 | Power transformer 1, comprising seven indicators such as oil temperature and useful load |
| ETTh2 | Electricity | 1 hour | 14,400 | 7 | 6:2:2 | Power transformer 2, comprising seven indicators such as oil temperature and useful load |
| ETTm1 | Electricity | 15 mins | 57,600 | 7 | 6:2:2 | Power transformer 1, comprising seven indicators such as oil temperature and useful load |
| ETTm2 | Electricity | 15 mins | 57,600 | 7 | 6:2:2 | Power transformer 2, comprising seven indicators such as oil temperature and useful load |
| Weather | Environment | 10 mins | 52,696 | 21 | 7:1:2 | Recorded every for the whole year 2020, which contains 21 meteorological indicators |
| Electricity | Electricity | 1 hour | 26,304 | 321 | 7:1:2 | Electricity records the electricity consumption in kWh every 1 hour from 2012 to 2014 |
| Solar | Energy | 10 mins | 52,560 | 137 | 6:2:2 | Solar production records collected from 137 PV plants in Alabama |
| Traffic | Traffic | 1 hour | 17,544 | 862 | 7:1:2 | Road occupancy rates measured by 862 sensors on San Francisco Bay area freeways |

# B Related Works

Time series forecasting (TSF) predicts future observations based on historical observations. TSF methods are mainly categorized into four distinct approaches: (1) statistical learning-based methods, (2) machine learning-based methods, (3) deep learning-based methods, and (4) foundation methods.

Early TSF methods primarily rely on statistical learning approaches such as ARIMA [Box and Pierce, 1970], ETS [Hyndman et al., 2008], and VAR [Godahewa et al., 2021]. With advancements in machine learning, methods like XGBoost [Chen and Guestrin, 2016], Random Forests [Breiman, 2001], and LightGBM [Ke et al., 2017] gain popularity for handling nonlinear patterns. However, these methods still require manual feature engineering and model design [Pan et al., 2023, Wu et al., 2024b, Lu et al., 2024, Li et al., 2025d, Fu et al., 2025, Lu et al., 2025, Li et al., 2025e, Zhou et al., 2025]. Leveraging the representation learning of deep neural networks (DNNs) [Cheng et al., 2023, Qiu et al., 2025d, Li et al., 2025f, Huang et al., 2025c, Yang et al., 2024, Lu et al., 2023, Wang et al., 2025f], many deep learning-based methods emerge. TimesNet [Wu et al., 2023b] and SegRNN [Lin et al., 2023] model time series as vector sequences, using CNNs or RNNs to capture temporal dependencies. Additionally, Transformer architectures, including DUET [Qiu et al., 2025a], Informer [Zhou et al., 2021], DAG [Qiu et al., 2025e], FEDformer [Zhou et al., 2022], Triformer [Cirstea et al., 2022], and PatchTST [Nie et al., 2023], capture complex relationships between time points more accurately, significantly improving forecasting performance. MLP-based methods, including Hdmixer [Huang et al., 2024], SparseTSF [Lin et al., 2024b], CycleNet [Lin et al., 2024a], APN [Liu et al., 2025d], SRSNet [Wu et al., 2025b], NLinear [Zeng et al., 2023], and DLinear [Zeng et al., 2023], adopt simpler architectures with fewer parameters but still achieve highly competitive forecasting accuracy.

However, many of these methods struggle with generalization across domains due to their reliance on domain-specific data [Li et al., 2025b]. To address this, foundation methods are proposed, categorized into LLM-based methods and time series pre-trained methods. LLM-based methods [Zhou et al., 2023, Jin et al., 2024, Liu et al., 2024a, Pan et al., 2024] leverage the strong representational capacity and sequential modeling capability of LLMs to capture complex patterns for time series modeling. Time series pre-trained methods [Liu et al., 2024b, Gao et al., 2024, Goswami et al., 2024, Das et al., 2024] focus on pre-training over multi-domain time series data, enabling the method to learn domain-agnostic features that are transferable across various applications. This strategy not only enhances performance on specific tasks but also provides greater flexibility when adapting to new datasets or scenarios.

# C  Theoretical Proofs

In this section, we provide a theoretical analysis to explain why the proposed DBLoss is more effective than the conventional MSE loss in time series forecasting.

Motivated by the success of recent methods such as DLinear Zeng et al. [2023], DUET Qiu et al. [2025a], TimeMixer Wang et al. [2024], and xPatch Stitsyuk and Choi [2025], which model time series by decomposing them into trend and seasonal components, achieving excellent performance, we assume that the trend and seasonal components are highly independent..

## C.1  Problem Formulation

Let the original time series be denoted as $y_t$, which can be decomposed into a trend component $T_t$ and a seasonal component $S_t$:

$$y_t = T_t + S_t. \tag{4}$$

Similarly, the model prediction $\hat{y}_t$ can be expressed as:

$$\hat{y}_t = \hat{T}_t + \hat{S}_t. \tag{5}$$

## C.2  Analysis of MSE Loss

Under this setting, the Mean Squared Error (MSE) can be expanded as:

$$
\begin{aligned}
L_{\text{MSE}} = \|y_t - \hat{y}_t\|_2^2 &= \|(T_t + S_t) - (\hat{T}_t + \hat{S}_t)\|_2^2 \\
&= \|(T_t - \hat{T}_t) + (S_t - \hat{S}_t)\|_2^2 \\
&= \|T_t - \hat{T}_t\|_2^2 + \|S_t - \hat{S}_t\|_2^2 + 2 \cdot (T_t - \hat{T}_t)(S_t - \hat{S}_t).
\end{aligned}
\tag{6}
$$

The key part from MSE lies in the cross term $2 \cdot (T_t - \hat{T}_t)(S_t - \hat{S}_t)$. Our assumption is that the trend and seasonal components are highly independent.However, this cross term introduces an interaction between them, potentially making it difficult for the model to optimize the two components independently, which can degrade the overall prediction performance. For instance, if the trend component is poorly predicted while the seasonal component is well captured, the interaction term can still yield a large negative value of $2 \cdot (T_t - \hat{T}_t)(S_t - \hat{S}_t)$, disproportionately affecting the total loss.

## C.3  Gradient Analysis of MSE

We further analyze the MSE loss from the perspective of gradient propagation, and reveal MSE loss being unable to independently consider these two components during the optimization process. According to the chain rule:

$$
\begin{aligned}
\nabla_{\boldsymbol{\Theta}} L_t &= 2 \cdot [(T_t - \hat{T}_t) + (S_t - \hat{S}_t)] \cdot \nabla_{\boldsymbol{\Theta}}(-\hat{T}_t - \hat{S}_t) \\
&= -2 \cdot [(T_t - \hat{T}_t) + (S_t - \hat{S}_t)] \cdot (\nabla_{\boldsymbol{\Theta}}\hat{T}_t + \nabla_{\boldsymbol{\Theta}}\hat{S}_t).
\end{aligned}
\tag{7}
$$

Let the Jacobians of the trend and seasonal components be:

$$\mathbf{J}_T := \nabla_{\boldsymbol{\Theta}}\hat{T}_t, \quad \mathbf{J}_S := \nabla_{\boldsymbol{\Theta}}\hat{S}_t. \tag{8}$$

Then the gradient can be expressed as:

$$\nabla_{\boldsymbol{\Theta}} L_t = -2 \cdot \left[(T_t - \hat{T}_t)\mathbf{J}_T + (T_t - \hat{T}_t)\mathbf{J}_S + (S_t - \hat{S}_t)\mathbf{J}_T + (S_t - \hat{S}_t)\mathbf{J}_S\right]. \tag{9}$$

We decompose this gradient into two parts:

**(1) Ideal Decoupled Term**

$$\mathbf{G}_{\text{ideal}} = -2 \cdot \left[(T_t - \hat{T}_t)\mathbf{J}_T + (S_t - \hat{S}_t)\mathbf{J}_S\right], \tag{10}$$

which represents the desired independent optimization of the two components.

**(2) Coupled Term**

$$\mathbf{G}_{\text{coupling}} = -2 \cdot \left[ (T_t - \hat{T}_t)\mathbf{J}_S + (S_t - \hat{S}_t)\mathbf{J}_T \right]. \tag{11}$$

The coupled term $\mathbf{G}_{\text{coupling}}$ causes mutual interference: the trend error influences the seasonal optimization and vice versa. As long as $T_t \neq \hat{T}_t$ or $S_t \neq \hat{S}_t$ (i.e., the model has not converged), $\mathbf{G}_{\text{coupling}} \neq \mathbf{0}$, meaning that the optimization of one component will inevitably affect the other.

### C.4   Analysis of DBLoss

$$L_{\text{DB}} = \beta \cdot \|\hat{S}_t - S_t\|_2^2 + (1 - \beta) \cdot \|\hat{T}_t - T_t\|_1, \tag{12}$$

where $\beta$ is hyperparameters controlling the relative weights of the trend and seasonal components.

Unlike MSE, DBLoss explicitly separates the optimization of the two components, thus removing the coupling term $\mathbf{G}_{\text{coupling}}$ from the gradient computation. Furthermore, by adjusting the coefficients or using different distance norms, one can precisely control the loss scale for each component, enabling targeted learning and better modeling of both parts.

## D   Visualization

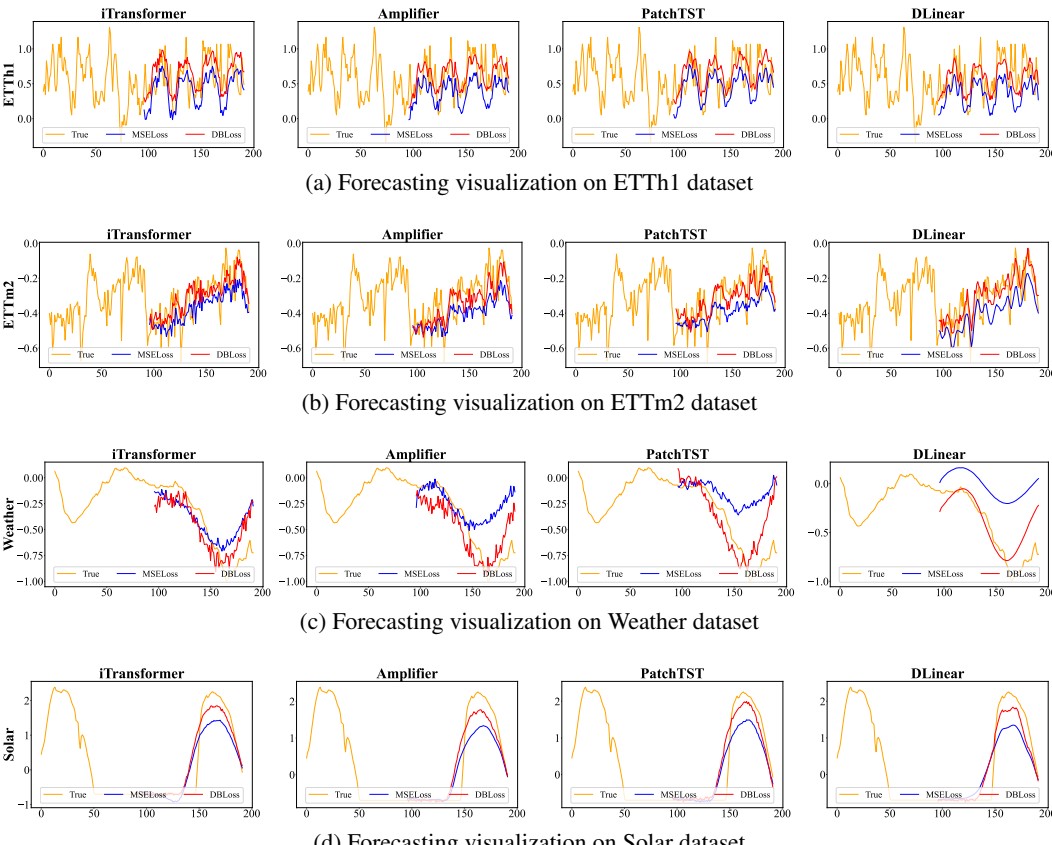

(a) Forecasting visualization on ETTh1 dataset

(b) Forecasting visualization on ETTm2 dataset

(c) Forecasting visualization on Weather dataset

(d) Forecasting visualization on Solar dataset

Figure 6: Forecasting visualization comparing DBLoss and MSE loss as objective functions.

# E  Comparison with Other Loss Functions

Table 6: Comparison between the proposed DBLoss and other loss functions. The model is DLinear and we report the result of ETTh2. The best results are highlighted in **bold**, and the second-best results are highlighted in underline. The standard deviation of methods calculated through 5 random seeds are also reported.

| Dataset | | ETTh2 | | | | |
|---|---|---|---|---|---|---|
| Forecast horizon | | 96 | 192 | 336 | 720 | Avg |
| Ori | MSE | $0.300_{\pm 0.002}$ | $0.387_{\pm 0.001}$ | $0.490_{\pm 0.001}$ | $0.704_{\pm 0.004}$ | $0.470_{\pm 0.001}$ |
| | MAE | $0.364_{\pm 0.002}$ | $0.423_{\pm 0.002}$ | $0.487_{\pm 0.002}$ | $0.597_{\pm 0.001}$ | $0.468_{\pm 0.002}$ |
| TILDE-Q | MSE | $0.287_{\pm 0.002}$ | $0.362_{\pm 0.002}$ | $0.425_{\pm 0.002}$ | $0.599_{\pm 0.001}$ | $0.418_{\pm 0.002}$ |
| (2022) | MAE | $0.345_{\pm 0.004}$ | $0.395_{\pm 0.001}$ | $0.445_{\pm 0.002}$ | $0.551_{\pm 0.002}$ | $0.434_{\pm 0.001}$ |
| FreDF | MSE | $\underline{0.284}_{\pm 0.001}$ | $0.362_{\pm 0.001}$ | $0.420_{\pm 0.002}$ | $\underline{0.587}_{\pm 0.002}$ | $\underline{0.413}_{\pm 0.001}$ |
| (2025e) | MAE | $\mathbf{0.342}_{\pm 0.003}$ | $0.396_{\pm 0.002}$ | $0.445_{\pm 0.004}$ | $\underline{0.546}_{\pm 0.003}$ | $0.432_{\pm 0.003}$ |
| PSLoss | MSE | $\mathbf{0.283}_{\pm 0.002}$ | $\underline{0.358}_{\pm 0.002}$ | $\underline{0.411}_{\pm 0.002}$ | $0.614_{\pm 0.003}$ | $0.417_{\pm 0.002}$ |
| (2025) | MAE | $0.343_{\pm 0.003}$ | $\underline{0.393}_{\pm 0.004}$ | $\underline{0.434}_{\pm 0.002}$ | $0.549_{\pm 0.003}$ | $\underline{0.430}_{\pm 0.002}$ |
| DBLoss | MSE | $\underline{0.284}_{\pm 0.002}$ | $\mathbf{0.357}_{\pm 0.003}$ | $\mathbf{0.407}_{\pm 0.002}$ | $\mathbf{0.586}_{\pm 0.001}$ | $\mathbf{0.409}_{\pm 0.001}$ |
| (Ours) | MAE | $\mathbf{0.342}_{\pm 0.001}$ | $\mathbf{0.390}_{\pm 0.001}$ | $\mathbf{0.430}_{\pm 0.001}$ | $\mathbf{0.533}_{\pm 0.004}$ | $\mathbf{0.424}_{\pm 0.002}$ |

Table 7: Comparison between the proposed DBLoss and other loss functions. The model is DLinear and we report the result of ETTm1. The best results are highlighted in **bold**, and the second-best results are highlighted in underline. The standard deviation of methods calculated through 5 random seeds are also reported.

| Dataset | | ETTm1 | | | | |
|---|---|---|---|---|---|---|
| Forecast horizon | | 96 | 192 | 336 | 720 | Avg |
| Ori | MSE | $0.300_{\pm 0.003}$ | $0.336_{\pm 0.002}$ | $0.367_{\pm 0.002}$ | $0.419_{\pm 0.003}$ | $0.356_{\pm 0.005}$ |
| | MAE | $0.345_{\pm 0.002}$ | $0.366_{\pm 0.002}$ | $0.386_{\pm 0.003}$ | $0.416_{\pm 0.003}$ | $0.378_{\pm 0.003}$ |
| TILDE-Q | MSE | $0.302_{\pm 0.002}$ | $0.336_{\pm 0.002}$ | $0.371_{\pm 0.002}$ | $0.425_{\pm 0.003}$ | $0.359_{\pm 0.003}$ |
| (2022) | MAE | $0.342_{\pm 0.005}$ | $0.362_{\pm 0.003}$ | $0.386_{\pm 0.002}$ | $0.417_{\pm 0.002}$ | $0.377_{\pm 0.002}$ |
| FreDF | MSE | $0.302_{\pm 0.003}$ | $0.333_{\pm 0.004}$ | $0.363_{\pm 0.001}$ | $\mathbf{0.415}_{\pm 0.001}$ | $0.353_{\pm 0.003}$ |
| (2025e) | MAE | $0.344_{\pm 0.001}$ | $0.363_{\pm 0.001}$ | $0.381_{\pm 0.003}$ | $\underline{0.411}_{\pm 0.002}$ | $0.375_{\pm 0.002}$ |
| PSLoss | MSE | $\underline{0.296}_{\pm 0.001}$ | $\underline{0.332}_{\pm 0.003}$ | $0.361_{\pm 0.001}$ | $0.416_{\pm 0.004}$ | $\mathbf{0.351}_{\pm 0.001}$ |
| (2025) | MAE | $\underline{0.339}_{\pm 0.002}$ | $\underline{0.361}_{\pm 0.001}$ | $\underline{0.380}_{\pm 0.002}$ | $0.413_{\pm 0.001}$ | $\underline{0.373}_{\pm 0.001}$ |
| DBLoss | MSE | $\mathbf{0.295}_{\pm 0.001}$ | $\mathbf{0.331}_{\pm 0.002}$ | $0.361_{\pm 0.002}$ | $\mathbf{0.415}_{\pm 0.001}$ | $\mathbf{0.351}_{\pm 0.002}$ |
| (Ours) | MAE | $\mathbf{0.337}_{\pm 0.001}$ | $\mathbf{0.358}_{\pm 0.001}$ | $\mathbf{0.378}_{\pm 0.001}$ | $\mathbf{0.409}_{\pm 0.001}$ | $\mathbf{0.370}_{\pm 0.002}$ |

Table 8: Comparison between the proposed DBLoss and other loss functions. The model is DLinear and we report the result of Traffic. The best results are highlighted in **bold**, and the second-best results are highlighted in underline. The standard deviation of methods calculated through 5 random seeds are also reported.

| Dataset | | Traffic | | | | |
|---|---|---|---|---|---|---|
| Forecast horizon | | 96 | 192 | 336 | 720 | Avg |
| Ori | MSE | $\mathbf{0.395}_{\pm 0.001}$ | $\mathbf{0.407}_{\pm 0.001}$ | $0.417_{\pm 0.001}$ | $0.454_{\pm 0.003}$ | $\underline{0.418}_{\pm 0.002}$ |
| | MAE | $0.275_{\pm 0.002}$ | $0.280_{\pm 0.001}$ | $0.286_{\pm 0.001}$ | $0.308_{\pm 0.001}$ | $0.287_{\pm 0.002}$ |
| TILDE-Q | MSE | $0.416_{\pm 0.003}$ | $0.422_{\pm 0.004}$ | $0.423_{\pm 0.001}$ | $0.461_{\pm 0.002}$ | $0.431_{\pm 0.003}$ |
| (2022) | MAE | $0.294_{\pm 0.002}$ | $0.296_{\pm 0.001}$ | $0.293_{\pm 0.002}$ | $0.316_{\pm 0.003}$ | $0.300_{\pm 0.002}$ |
| FreDF | MSE | $0.398_{\pm 0.001}$ | $0.408_{\pm 0.004}$ | $\underline{0.416}_{\pm 0.002}$ | $\underline{0.452}_{\pm 0.001}$ | $0.419_{\pm 0.001}$ |
| (2025e) | MAE | $\mathbf{0.270}_{\pm 0.001}$ | $0.275_{\pm 0.003}$ | $0.280_{\pm 0.002}$ | $0.302_{\pm 0.002}$ | $0.282_{\pm 0.001}$ |
| PSLoss | MSE | $0.398_{\pm 0.001}$ | $0.408_{\pm 0.001}$ | $\underline{0.416}_{\pm 0.003}$ | $\underline{0.452}_{\pm 0.005}$ | $0.419_{\pm 0.002}$ |
| (2025) | MAE | $\mathbf{0.270}_{\pm 0.001}$ | $\mathbf{0.274}_{\pm 0.001}$ | $\mathbf{0.279}_{\pm 0.001}$ | $\underline{0.299}_{\pm 0.003}$ | $\underline{0.281}_{\pm 0.002}$ |
| DBLoss | MSE | $\underline{0.396}_{\pm 0.001}$ | $\mathbf{0.407}_{\pm 0.001}$ | $\mathbf{0.415}_{\pm 0.001}$ | $\mathbf{0.449}_{\pm 0.005}$ | $\mathbf{0.417}_{\pm 0.002}$ |
| (Ours) | MAE | $\mathbf{0.270}_{\pm 0.001}$ | $\mathbf{0.274}_{\pm 0.001}$ | $\mathbf{0.279}_{\pm 0.001}$ | $\mathbf{0.298}_{\pm 0.003}$ | $\mathbf{0.280}_{\pm 0.002}$ |

## F Efficiency

Table 9 presents the epoch-level training times (in seconds) of PatchTST when using DBLoss and MSE across different datasets. The results show the average values for the four forecasting horizons of each dataset, with the same parameters, where only MSE is replaced by DBLoss. Based on the experimental results in the table, we can observe that DBLoss does lead to an increase in training time compared to MSE, but this increase is not significant. As the dataset size grows, the time difference becomes even more negligible.

Table 9: Comparison of average epoch-level training times (in seconds) between DBLoss and MSE using PatchTST across four forecasting horizons on different datasets.

| Train Time | ETTh1 | ETTh2 | ETTm1 | ETTm2 | Solar | Weather | Electricity | Traffic |
|---|---|---|---|---|---|---|---|---|
| MSE | 2.36 | 2.37 | 14.45 | 14.39 | 183.11 | 36.07 | 258.47 | 1035.77 |
| DBLoss | 3.11 | 3.37 | 15.93 | 15.73 | 186.31 | 37.23 | 260.85 | 1039.67 |

## G Full Results on Time Series Foundation Models

Table 10 presents the results of foundation models under the 5% few-shot setting. It reports both MSE and MAE across different forecasting horizons $F \in 96, 192, 336, 720$. The baseline parameters are kept consistent with those used in TSFM-Bench [Li et al., 2025c]. The best results are highlighted in **bold**.

## H Limitations

**Potential limitations** The DBLoss demonstrates its efficacy in TSF scenarios. However, there are several potential limitations of DBLoss that warrant discussion here:

- **Lack of Automated Hyperparameter Tuning Strategy:** The proposed method involves two critical hyperparameters: the score weight $\beta$ for the weighted loss and the smoothing factor $\alpha$ for Exponential Moving Average (EMA) decomposition. Specifically, a larger $\beta$ increases the proportion of the seasonal component in the loss calculation, while a smaller $\alpha$ results in stronger smoothing, making the trend smoother and the seasonal component more prominent. However, the current research has not yet proposed an automated strategy to optimize the selection of these two parameters. The absence of a systematic tuning approach may still limit the model's performance improvement and generalization capability. Therefore, developing an efficient automated hyperparameter tuning mechanism to adaptively determine the optimal parameter combination is an important direction for future research.

Table 10: Foundation models results in the 5% few-shot setting. The table reports MSE and MAE for different forecasting lengths $F \in \{96, 192, 336, 720\}$. The parameters for the baselines are kept consistent with those of TSFM-Bench [Li et al., 2025c]. The better results are highlighted in **bold**.

| Model | | GPT4TS | | | | CALF | | | | TTM | | | | UniTS | | | |
|---|---|---|---|---|---|---|---|---|---|---|---|---|---|---|---|---|---|
| Loss | | Ori | | DBLoss | | Ori | | DBLoss | | Ori | | DBLoss | | Ori | | DBLoss | |
| Metric | | MSE | MAE | MSE | MAE | MSE | MAE | MSE | MAE | MSE | MAE | MSE | MAE | MSE | MAE | MSE | MAE |
| ETTh1 | 96 | **0.438** | **0.445** | 0.439 | 0.446 | 0.405 | 0.426 | **0.401** | **0.422** | 0.363 | 0.392 | **0.361** | **0.390** | **0.381** | 0.394 | **0.381** | **0.393** |
| | 192 | 0.460 | 0.458 | **0.455** | **0.456** | 0.428 | 0.442 | **0.423** | **0.436** | 0.391 | 0.409 | **0.387** | **0.405** | 0.421 | 0.428 | **0.405** | **0.424** |
| | 336 | 0.462 | 0.467 | **0.449** | **0.460** | 0.443 | 0.454 | **0.437** | **0.447** | 0.411 | 0.429 | **0.404** | **0.422** | 0.443 | 0.437 | **0.429** | **0.429** |
| | 720 | 0.509 | 0.511 | **0.470** | **0.486** | 0.495 | 0.494 | **0.472** | **0.479** | 0.453 | 0.471 | **0.429** | **0.448** | 0.498 | 0.475 | **0.486** | **0.463** |
| | Avg | 0.467 | 0.470 | **0.453** | **0.462** | 0.443 | 0.454 | **0.433** | **0.446** | 0.405 | 0.425 | **0.395** | **0.417** | 0.436 | 0.434 | **0.425** | **0.427** |
| ETTh2 | 96 | 0.329 | 0.380 | **0.323** | **0.371** | **0.302** | **0.362** | 0.308 | 0.363 | 0.271 | 0.329 | **0.270** | **0.328** | 0.305 | 0.353 | **0.299** | **0.346** |
| | 192 | 0.368 | 0.406 | **0.364** | **0.399** | 0.385 | 0.400 | **0.383** | **0.397** | 0.339 | **0.373** | **0.329** | 0.374 | 0.369 | 0.403 | **0.357** | **0.390** |
| | 336 | 0.378 | 0.421 | **0.374** | **0.412** | 0.387 | 0.418 | **0.375** | **0.412** | 0.372 | 0.401 | **0.346** | **0.392** | 0.388 | 0.412 | **0.361** | **0.399** |
| | 720 | 0.418 | 0.450 | **0.412** | **0.442** | 0.416 | 0.449 | **0.407** | **0.445** | 0.385 | 0.428 | **0.382** | **0.419** | 0.425 | 0.451 | **0.410** | **0.439** |
| | Avg | 0.373 | 0.414 | **0.368** | **0.406** | 0.373 | 0.407 | **0.368** | **0.404** | 0.342 | 0.383 | **0.332** | **0.378** | 0.372 | 0.405 | **0.357** | **0.393** |
| ETTm1 | 96 | 0.343 | 0.379 | **0.330** | **0.368** | 0.317 | 0.366 | **0.299** | **0.347** | 0.299 | 0.343 | **0.294** | **0.337** | 0.313 | 0.363 | **0.300** | **0.349** |
| | 192 | 0.375 | 0.398 | **0.361** | **0.387** | 0.346 | 0.380 | **0.337** | **0.369** | 0.341 | 0.367 | **0.340** | **0.365** | 0.357 | 0.390 | **0.338** | **0.373** |
| | 336 | 0.394 | 0.406 | **0.384** | **0.398** | 0.385 | 0.405 | **0.371** | **0.391** | 0.365 | 0.381 | **0.363** | **0.379** | 0.381 | 0.405 | **0.370** | **0.392** |
| | 720 | 0.440 | 0.434 | **0.432** | **0.425** | 0.439 | 0.433 | **0.427** | **0.420** | 0.420 | 0.412 | **0.419** | **0.409** | 0.457 | 0.448 | **0.438** | **0.428** |
| | Avg | 0.388 | 0.404 | **0.377** | **0.394** | 0.372 | 0.396 | **0.358** | **0.382** | 0.356 | 0.376 | **0.354** | **0.372** | 0.377 | 0.402 | **0.362** | **0.386** |
| ETTm2 | 96 | 0.190 | 0.279 | **0.181** | **0.266** | 0.180 | 0.272 | **0.170** | **0.258** | 0.164 | 0.250 | **0.162** | **0.244** | 0.188 | 0.278 | **0.173** | **0.255** |
| | 192 | 0.241 | 0.312 | **0.231** | **0.300** | 0.237 | 0.310 | **0.228** | **0.295** | 0.222 | 0.290 | 0.224 | **0.287** | 0.255 | 0.317 | **0.247** | **0.302** |
| | 336 | 0.296 | 0.349 | **0.281** | **0.330** | 0.295 | 0.348 | **0.277** | **0.328** | 0.282 | 0.330 | **0.278** | **0.323** | 0.321 | 0.366 | **0.285** | **0.331** |
| | 720 | 0.385 | 0.401 | **0.371** | **0.384** | 0.372 | 0.397 | **0.363** | **0.382** | 0.364 | 0.381 | **0.363** | **0.376** | 0.404 | 0.415 | **0.374** | **0.392** |
| | Avg | 0.278 | 0.335 | **0.266** | **0.320** | 0.271 | 0.332 | **0.259** | **0.316** | 0.258 | 0.313 | **0.257** | **0.308** | 0.292 | 0.344 | **0.270** | **0.320** |
| Solar | 96 | 0.253 | 0.326 | **0.243** | **0.267** | **0.203** | **0.274** | 0.238 | 0.295 | **0.201** | 0.254 | **0.201** | **0.244** | **0.186** | 0.244 | **0.186** | **0.226** |
| | 192 | 0.266 | 0.336 | **0.252** | **0.281** | **0.224** | **0.290** | 0.243 | 0.294 | 0.225 | 0.270 | 0.235 | **0.267** | **0.198** | 0.255 | 0.206 | **0.241** |
| | 336 | 0.262 | 0.341 | **0.260** | **0.284** | **0.243** | **0.308** | 0.252 | 0.303 | 0.222 | 0.274 | 0.231 | **0.271** | **0.208** | 0.259 | 0.222 | **0.254** |
| | 720 | 0.265 | 0.335 | **0.261** | **0.285** | **0.247** | **0.314** | 0.251 | 0.306 | **0.226** | **0.277** | 0.229 | 0.283 | **0.231** | 0.284 | 0.243 | **0.265** |
| | Avg | 0.262 | 0.335 | **0.254** | **0.279** | **0.229** | **0.297** | 0.246 | 0.300 | **0.219** | 0.269 | 0.224 | **0.266** | **0.206** | 0.261 | 0.214 | **0.246** |
| Weather | 96 | 0.187 | 0.244 | **0.181** | **0.234** | 0.163 | 0.217 | **0.162** | **0.210** | **0.147** | 0.195 | 0.148 | **0.191** | **0.154** | 0.206 | 0.156 | **0.198** |
| | 192 | 0.225 | 0.274 | **0.220** | **0.265** | 0.206 | 0.253 | **0.205** | **0.250** | **0.194** | 0.238 | **0.194** | **0.233** | **0.199** | 0.248 | **0.199** | **0.237** |
| | 336 | 0.268 | 0.304 | **0.265** | **0.297** | 0.260 | 0.297 | **0.257** | **0.292** | **0.244** | 0.277 | 0.243 | **0.272** | **0.248** | 0.285 | **0.248** | **0.275** |
| | 720 | 0.330 | 0.348 | **0.327** | **0.340** | **0.322** | 0.339 | 0.322 | **0.338** | **0.314** | 0.329 | 0.316 | **0.327** | **0.320** | 0.337 | **0.320** | **0.329** |
| | Avg | 0.253 | 0.293 | **0.248** | **0.284** | 0.238 | 0.277 | **0.236** | **0.272** | **0.225** | 0.260 | **0.225** | **0.256** | **0.230** | 0.269 | 0.231 | **0.260** |
| Electricity | 96 | **0.178** | 0.294 | 0.179 | **0.286** | 0.141 | 0.240 | **0.140** | **0.236** | **0.146** | 0.246 | **0.146** | **0.245** | **0.150** | 0.249 | 0.152 | **0.248** |
| | 192 | **0.192** | 0.306 | **0.192** | **0.300** | 0.156 | 0.254 | **0.154** | **0.250** | **0.165** | 0.264 | **0.165** | **0.262** | **0.167** | 0.264 | 0.168 | **0.263** |
| | 336 | **0.208** | 0.318 | **0.208** | **0.310** | 0.174 | 0.271 | **0.174** | **0.268** | 0.181 | 0.281 | **0.180** | **0.278** | **0.181** | 0.277 | 0.182 | **0.276** |
| | 720 | **0.248** | 0.348 | **0.248** | **0.339** | 0.216 | 0.306 | **0.214** | **0.302** | **0.223** | 0.315 | **0.223** | **0.313** | **0.220** | 0.309 | 0.224 | **0.310** |
| | Avg | **0.207** | 0.317 | **0.207** | **0.309** | 0.172 | 0.268 | **0.171** | **0.264** | 0.179 | 0.277 | **0.178** | **0.274** | **0.180** | 0.275 | 0.181 | **0.274** |
| Traffic | 96 | 0.411 | 0.300 | **0.408** | **0.286** | 0.406 | 0.298 | **0.405** | **0.290** | 0.448 | 0.324 | **0.445** | **0.322** | 0.401 | 0.278 | **0.400** | **0.272** |
| | 192 | 0.422 | 0.304 | **0.416** | **0.289** | 0.423 | 0.309 | **0.420** | **0.301** | 0.466 | 0.330 | **0.463** | **0.329** | 0.414 | 0.284 | **0.412** | **0.278** |
| | 336 | 0.432 | 0.308 | **0.426** | **0.293** | 0.436 | 0.317 | **0.432** | **0.310** | 0.491 | 0.345 | **0.487** | **0.343** | 0.421 | 0.290 | **0.417** | **0.280** |
| | 720 | 0.468 | 0.325 | **0.463** | **0.311** | 0.477 | 0.340 | **0.476** | **0.335** | 0.533 | 0.365 | **0.527** | **0.361** | 0.452 | 0.305 | **0.451** | **0.298** |
| | Avg | 0.433 | 0.309 | **0.428** | **0.295** | 0.435 | 0.316 | **0.433** | **0.309** | 0.484 | 0.341 | **0.481** | **0.339** | 0.422 | 0.289 | **0.420** | **0.282** |

