# OpenReview forum: "DBLoss: Decomposition-based Loss Function for Time Series Forecasting"
_NeurIPS.cc/2025/Conference — NeurIPS 2025 poster_

### Official Review · Reviewer_4VN9 · 2025-06-24

**Clarity:** 2
**Significance:** 3
**Originality:** 2
**Rating:** 4
**Confidence:** 3

**Summary:**

The paper introduces DBLoss, decomposition-based loss function designed to improve time series forecasting performance by modeling seasonality and trend components within the forecasting horizon.  DBLoss employs EMA, exponential moving average, to seperate into each components. Experiments demonstrate that the DBLoss consistently outperforms traditional MSE loss by comparing SOTA models trained by each loss.

**Questions:**

As weaknesses.

**Ethical Concerns:**

["NO or VERY MINOR ethics concerns only"]

**Final Justification:**

They gave the detailed response and the additional experimental results provided for clarification. Incorporating this information into your paper would enhance the clarity of your explanations and strengthen the experimental support for your contributions. I will incorporate these response into the evaluation score under the assumption that these revisions will be included in your final version.

**Limitations:**

As weaknesses.

**Paper Formatting Concerns:**

No concerns.

**Quality:**

3

**Strengths And Weaknesses:**

Strengths:
Identifies a limitation of MSE loss function for TSF, making the problem clearly. Proposed DBLoss is easily integrable into various forecasting models without significant increasing computational costs, which can lead to high potential for practical adoption. Experiments are comprehensive, covering diverse datasets and various type of SOTA models. Several robustness experiments are shown, too.

Weaknesses:
1. Contrasting to strong empirical results, theoretical justification or analysis of why DBLoss improves forecasting performance is lacking. A clear theoretical explanation or analysis of empirical results is needed.
2. There should be a empirical or theoretical justification of why 2-norm is used for seasonality loss and 1-norm is used for trend loss.
3. The method relies on existing, well-known exponential moving average decomposition. Novelty of the methodological contribution is quite limited.
4. Computational cost analysis must be shown for adapting this loss. Empirical timing experiments would make it clearer.

---

> ### Author Rebuttal · Authors · 2025-07-30
>
> Dear Reviewer **4VN9**, thank you for providing your detailed and constructive feedback.
>
> **W1:** A clear theoretical explanation or analysis of empirical results is needed.
>
> Motivated by the success of methods such as Dlinear, DUET, TimeMixer, and xPatch, which model time series by decomposing them into trend and seasonal components, achieving excellent performance, we assume that the trend and seasonal components are highly independent.
>
> We represent the time series as $y_t$, which can be decomposed into a trend component $T_t$ and a seasonal component $S_t$, such that:
>
> $y_t = T_t + S_t$
>
> For the model’s prediction $\hat{y}_t$, we similarly have:
>
> $\hat{y}_t = \hat{T}_t + \hat{S}_t$
>
> Under this setting, the Mean Squared Error (MSE) can be expanded as:
>
> $$L _{MSE} = ||y _t - \hat{y} _t|| _2 ^2 = ||(T _t + S _t) - (\hat{T} _t + \hat{S} _t)||_2 ^2 = ||(T _t - \hat{T}_t) + (S _t - \hat{S} _t)||_2^2= ||T _t - \hat{T}_t|| _2^2 + ||S _t - \hat{S}_t||_2 ^2 + 2 \cdot (T_t - \hat{T}_t) \cdot (S_t - \hat{S}_t)$$
>
> The key part from MSE lies in the cross term $2 \cdot (T_t - \hat{T}_t) \cdot (S_t - \hat{S}_t)$. Our assumption is that the trend and seasonal components are highly independent.However, this cross term introduces an interaction between them, potentially making it difficult for the model to optimize the two components independently, which can degrade the overall prediction performance. For instance, if the trend component is poorly predicted while the seasonal component is well captured, the interaction term can still yield a large negative value of $2 \cdot (T_t - \hat{T}_t) \cdot (S_t - \hat{S}_t)$, disproportionately affecting the total loss.
>
> Furthermore, we analyze $L_{MSE}$ from the perspective of differentiation, and reveal MSE loss function being unable to independently consider these two components during the optimization process:
>
> According to the chain rule:
>
> $\nabla _{\boldsymbol{\Theta}} L _t = 2 \cdot [(T _t - \hat{T}_t) + (S _t - \hat{S}_t)] \cdot \nabla _{\boldsymbol{\Theta}} (-\hat{T}_t - \hat{S}_t) = -2 \cdot [(T _t - \hat{T}_t) + (S _t - \hat{S}_t)] \cdot \left( \nabla _{\boldsymbol{\Theta}} \hat{T}_t + \nabla _{\boldsymbol{\Theta}} \hat{S}_t \right)$
>
> Using Jacobian matrix notation:
>
> - Jacobian of Trend: $$ \mathbf{J} _T := \nabla _{\boldsymbol{\Theta}} \hat{T} _t  $$
> - Jacobian of Seasonality: $$ \mathbf{J}_S := \nabla _{\boldsymbol{\Theta}} \hat{S}_t $$
>
> Then:
>
> $$ \nabla_{\boldsymbol{\Theta}} L_t = -2 \cdot [(T_t - \hat{T}_t) + (S_t - \hat{S}_t)] \cdot (\mathbf{J}_T + \mathbf{J}_S) $$
>
> $$ \nabla_{\boldsymbol{\Theta}} L_t = -2 \cdot \left[ (T_t - \hat{T}_t)(\mathbf{J}_T + \mathbf{J}_S) + (S_t - \hat{S}_t)(\mathbf{J}_T + \mathbf{J}_S) \right] $$
>
> Expanding the terms:
>
> $$ \nabla_{\boldsymbol{\Theta}} L_t = -2 \cdot \left[ (T_t - \hat{T}_t)\mathbf{J}_T + (T_t - \hat{T}_t)\mathbf{J}_S + (S_t - \hat{S}_t)\mathbf{J}_T + (S_t - \hat{S}_t)\mathbf{J}_S \right] $$
>
> We split this into two parts:
>
> 1、Ideal Decoupled Term
>
> $$ \mathbf{G}_{\text{ideal}} = -2 \cdot \left[ (T_t - \hat{T}_t)\mathbf{J}_T + (S_t - \hat{S}_t)\mathbf{J}_S \right] $$
>
> 2、Coupled Term
>
> $$ \mathbf{G}_{\text{coupling}} = -2 \cdot \left[ (T_t - \hat{T}_t)\mathbf{J}_S + (S_t - \hat{S}_t)\mathbf{J}_T \right] $$
>
> This term represents the problematic coupling:
>
> - The trend error influences the seasonality‑guided optimization.
> - The seasonal error influences the trend‑guided optimization.
>
> The full gradient is expressed as the sum of the ideal and coupled terms: $$ \nabla _{\boldsymbol{\Theta}} L _t = \mathbf{G} _{\text{ideal}} + \mathbf{G} _{\text{coupling}} $$
>
> As long as $T _t \neq \hat{T}_t $ or $S _t  \neq \hat{S} _t $ (i.e., the model has not fully converged), the coupled term $  \mathbf{G} _{\text{coupling}}  \neq \mathbf{0} $. Due to the presence of $ \mathbf{G} _{\text{coupling}} $, the MSE loss function being unable to independently consider these two components during the optimization process.
>
>
> Compared with MSE, DBLoss ($L _{DB} = \lambda_1 ||T _t - \hat{T} _t||_1 + \lambda_2 ||S _t - \hat{S}_t||_2^2$ , where $\lambda_1$ and $\lambda_2$ are hyperparameters controlling the weights of the trend and seasonal losses, respectively.) removes this cross term (or $  \mathbf{G} _{\text{coupling}} $), enabling the model to treat these two components independently throughout the optimization process.   Furthermore, by adjusting the coefficients or using different distance norms, one can precisely control the loss scale for each component, enabling targeted learning and better modeling of both parts.
>
>
> **W2:** There should be a empirical or theoretical justification of why 2-norm is used for seasonality loss and 1-norm is used for trend loss.
>
> Your suggestion is crucial for enhancing the clarity of our paper. Based on the experimental results, we chose to use the 2-norm for seasonality loss and the 1-norm for trend loss. Specifically, we tested the following combinations: 2-norm for both seasonality and trend losses (L2-L2), 1-norm for seasonality and 2-norm for trend loss (L1-L2), and 2-norm for seasonality and 1-norm for trend loss (DBLoss (L2-L1)). We found that the best performance was often achieved when using the 2-norm for seasonality loss and the 1-norm for trend loss.
>
>
>
> | Model         |      | iTransformer |       |      |       |             |       |     |      | TimeMixer |       |       |       |    |       |     |       |
> | ------------- | ---- | ------------ | ----- | ----------- | ----- | ----------- | ----- | --------------------- | ----- | --------- | ----- | ----------- | ----- | ----------- | ----- | --------------------- | ----- |
> | Loss Function |      | MSE          |       |L2-L2 |       | L2-L1 |       | DBLoss |       | MSE       |       | L2-L2 |       | L2-L1|       | DBLoss |       |
> | Metric        |      | MSE          | MAE   | MSE         | MAE   | MSE         | MAE   | MSE                   | MAE   | MSE       | MAE   | MSE         | MAE   | MSE         | MAE   | MSE                   | MAE   |
> | ETTh2         | Avg  | 0.370        | 0.403 | 0.368       | 0.401 | **0.362**       | 0.397 | 0.363                 | **0.395** | 0.349     | 0.397 | **0.344**       | **0.386** | 0.352       | 0.397 | 0.346                 | 0.387 |
> | ETTm2         | Avg  | 0.269        | 0.327 | 0.264       | 0.324 | 0.265       | 0.324 | **0.262**                 | **0.317** | 0.257     | 0.318 | 0.253       | 0.312 | 0.254       | 0.312 | **0.251**                 | **0.307** |
> | Weather       | Avg  | 0.232        | 0.270 | 0.231       | 0.268 | 0.234       | 0.270 | **0.229**                 | **0.261** | 0.226     | 0.264 | 0.225       | 0.259 | 0.224       | 0.259 | **0.222**                 | **0.254** |
>
>
>
> | Model         |      | PatchTST |       |    |       |     |       |    |       | DLinear |       |    |       |    |       |   |     |
> | ------------- | ---- | -------- | ----- | ----------- | ----- | ----------- | ----- | --------------------- | ----- | ------- | ----- | ----------- | ----- | ----------- | ----- | --------------------- | ----------- |
> | Loss Function |      | MSE      |       | L2-L2 |       | L2-L1 |       | DBLoss  |       | MSE     |       | L2-L2 |       | L2-L1 |       | DBLoss  |             |
> | Metric        |      | MSE      | MAE   | MSE         | MAE   | MSE         | MAE   | MSE                   | MAE   | MSE     | MAE   | MSE         | MAE   | MSE         | MAE   | MSE                   | MAE         |
> | ETTh2         | Avg  | 0.351    | 0.395 | **0.337**       | 0.386 | 0.339       | 0.385 | 0.339                 | **0.383** | 0.470   | 0.468 | 0.409       | 0.435 | 0.411       | 0.436 | **0.391**                 | **0.414**       |
> | ETTm2         | Avg  | 0.256    | 0.314 | 0.256       | 0.309 | 0.256       | 0.310 | **0.252**                 | **0.307** | 0.259   | 0.324 | 0.257       | 0.320 | **0.254**       | 0.319 | 0.256                 | **0.313**       |
> | Weather       | Avg  | 0.224    | 0.262 | **0.222**       | 0.252 | **0.222**       | **0.250** | 0.223                 | 0.252 | 0.242   | 0.293 | 0.241       | 0.288 | **0.238**       | **0.279** | 0.239                 | 0.280 |
>
> **W3:** The method relies on existing, well-known exponential moving average decomposition.
>
> Thank you for your valuable feedback. Indeed, we did not make an innovative contribution in terms of the decomposition method itself. Our main contribution lies in proposing a new loss paradigm that, through a reasonable decomposition approach, can enhance model performance. In fact, any decomposition method could be applied, but the quality of the decomposition directly affects the final performance. We adopted the currently advanced EMA decomposition method [1] with the aim of improving performance.
>
> [1] xPatch: Dual-Stream Time Series Forecasting with Exponential Seasonal-Trend Decomposition, AAAI, 2025
>
> **W4:** Empirical timing experiments would make it clearer.
>
> | Train Time  | ETTh1 | ETTh2 | ETTm1 | ETTm2 | Solar  | Weather | Electricity | Traffic |
> | -------------------------- | ----- | ----- | ----- | ----- | ------ | ------- | ----------- | ------- |
> | MSE                        | 2.36  | 2.37  | 14.45 | 14.39 | 183.11 | 36.07   | 258.47      | 1035.77 |
> | DBLoss                     | 3.11  | 3.37  | 15.93 | 15.73 | 186.31 | 37.23   | 260.85      | 1039.67 |
>
> The table presents the epoch-level training times (in seconds) when using DBLoss compared to MSE on different datasets. The results show the average values for the four prediction steps of each dataset, with the same parameters, where only MSE is replaced by DBLoss. Based on the experimental results in the table, we can observe that DBLoss does lead to an increase in training time compared to MSE, but this increase is not significant. As the dataset size grows, the time difference becomes even more negligible.

---

> > ### Comment · Reviewer_4VN9 · 2025-08-04
> >
> > Thank you for the detailed response and the additional experimental results provided for clarification. Incorporating this information into your paper would enhance the clarity of your explanations and strengthen the experimental support for your contributions. I will incorporate these response into the evaluation score under the assumption that these revisions will be included in your final version.

---

> ### Author Response · Authors · 2025-08-04
> **Dear Reviewer 4VN9, thank you for your valuable feedback and constructive suggestions**
>
> Dear Reviewer 4VN9, thank you for your valuable feedback and constructive suggestions. I appreciate your acknowledgment of the experimental results and your recommendation to incorporate this information into the paper. I will ensure that these revisions are included in the final version of the manuscript, thereby enhancing the clarity of the explanations and providing stronger experimental support for the contributions.
> ﻿
> Once again, thank you for your time and thoughtful evaluation.

---

### Official Review · Reviewer_PehU · 2025-06-25

**Clarity:** 3
**Significance:** 3
**Originality:** 2
**Rating:** 4
**Confidence:** 4

**Summary:**

This paper identifies a key limitation of the commonly used MSE loss in time-series forecasting. To address this limation, the authors propose DBLoss, a decomposition-based loss function. DBLoss works by applying EMA decomposition to both the model's prediction and the ground truth, extracting their seasonal and trend components. It then computes the losses for each component separately, L2 loss for seasonal component and L1 loss for trend component, and combined via a weighted sum, controlled by a tunable hyperparameter $\beta$. The method is model-agnostic and can be seamlessly applied to various time series forecasting models.

**Questions:**

1. Time series data inherently follows certain statistical principles and often comes with well-estabilished decomposition theories. However,  the authors do not discuss which theoretical assumptions or statistical interpretations justify the use of DBLoss. While the empirical results are strong, I believe that a theoretical foundation is essential --- especially when introducing a new loss function --- so that we can better understand when and why DBLoss should work. The paper would be strengthened by a deeper discussion of such theoretical grounding.
2. Follow question from Weakness 3 : Could the limited impact of the score weight $\beta$ be due to  the residual noise component remaining in both the trend and seasonal parts after decomposition?

**Ethical Concerns:**

["NO or VERY MINOR ethics concerns only"]

**Final Justification:**

The paper addresses a timely and important problem in time-series forecasting: the limitations of MSE loss in modeling trend and seasonality. The motivation is well-grounded, and the proposed DBLoss is model-agnostic, requiring no architectural changes for adoption. Extensive experiments across diverse datasets and models demonstrate consistent and meaningful performance improvements, including in zero-shot and few-shot scenarios, highlighting its broad applicability.

All initial concerns raised during the first review round have been adequately addressed in the rebuttal and discussion. In particular, the authors clarified the role of the score weight $\beta$ and its impact on component dominance, explained how DBLoss differs conceptually from simple reweighting of MSE, and discussed its robustness to residual/noise patterns. These clarifications, combined with the thorough empirical evidence, resolve my earlier reservations.

Given the clarity of the motivation, the strength of the experimental results, and the successful resolution of prior concerns, I now consider the contribution solid and impactful. I therefore lean toward a borderline accept recommendation.

**Limitations:**

yes

**Paper Formatting Concerns:**

.

**Quality:**

2

**Strengths And Weaknesses:**

Strengths :
1. The paper address a timely and important issue in time-series forecasting: the limitations of MSE loss in modeling trend and seasonality. The motivation is clearly articulated and well-grounded.
2. DBLoss is model-agnostic and can be easily incorporated into various deep learning forecasting models without requiring architectural changes.
3. Extensive experiments on diverse datasets and models demonstrate consistent performance improvements, including in zero-shot and few-shot scenarios.

Weakness :
1. In Figure 4, the sensitivity analysis of the score weight $\beta$ shows relatively small performance difference across a wide range of values. Although the authors acknowledge in Appendix D that the score weight $\beta$ has limited impact within a certain range, I believe this observation points to a more fundamental  issue: DBLoss may lack explicitly control over which component dominates the learning process. For instance, if a dataset has dominant seasonal pattern, the model may still overly focus on that, similar to implicit bias in MSE training.
2. While DBLoss decomposes the prediction and ground truth into seasonal and trend components, it ultimately sums the two losses and backpropagates through both. This raises the question: does it truly overcome the limitations of MSE, or is it just a reweighted version of the same distance based loss concept?
3. The method applies EMA for decomposition, which is a smoothing technique. However, it does not explicitly account for residuals or noise components after decomposition. It remains unclear how well DBLoss handles unpredictable or irregular patterns that are not capturd by trend or seasonality.

---

> ### Author Rebuttal · Authors · 2025-07-30
>
> We would like to sincerely thank **Reviewer PehU** for providing a detailed review and insightful comments. We have revised our paper accordingly.
>
> **W1&Q2:** The limited impact of the score weight.
>
> Dear Reviewer **PehU,** Thank you for your insightful feedback and constructive observations regarding our work. We truly appreciate your engagement with our study. Your comments prompted us to further reflect on our approach, and upon deeper investigation, we found that, as you mentioned, the parameter β did not adequately control the balance between the trend and seasonality components in the learning process.
>
> We believe this issue arose because we did not account for the differing value scales of the trend loss and season loss in our original approach. To address this, we introduced a modification in the loss function:
>
> $$\text{trend\\_loss} = \text{trend\\_loss} \times \frac{\text{season\\_loss}}{(\text{trend\\_loss} + 1e^{-8})}.\text{detach()}$$
>
> By introducing this equation, we normalize the trend loss by the relative scale of the season loss, ensuring that both components are numerically consistent, treated as a good initialization strategy. In particular, the use of `.detach()` ensures that the gradient of the season loss does not influence the computation of the trend loss during backpropagation. This means that the season loss is used solely for scaling purposes, without affecting the gradient updates for the trend loss, thus maintaining the balance and stability between the two components. Then we can use the β to further control the influence of trend and seasonality on the training process, which makes the importance of β more explict. We subsequently conducted a reanalysis of the sensitivity of β on the ETTh2 and Traffic datasets, and retested the results in Tables 1 using the modified version. We are pleased to report that this adjustment led to an improvement in DBLoss performance. **Due to character limitations, we will present the results (Table 1) during the rebuttal discussion.** Thank you for your valuable feedback.
>
> | |       |       |       |       |       | $\beta$ |       |       |       |       |       |
> | ---------- | ----- | ----- | ----- | ----- | ----- | ------- | ----- | ----- | ----- | ----- | ----- |
> |            | 0     | 0.1   | 0.2   | 0.3   | 0.4   | 0.5     | 0.6   | 0.7   | 0.8   | 0.9   | 1     |
> | ETTh2      | 0.332 | 0.281 | 0.281 | 0.282 | 0.283 | 0.284   | 0.286 | 0.291 | 0.300 | 0.314 | 0.340 |
> | Traffic    | 0.418 | 0.410 | 0.408 | 0.405 | 0.402 | 0.399   | 0.398 | 0.398 | 0.396 | 0.392 | 0.421 |
>
>  The impact of the hyperparameter ($\beta$) on ETTh2 and Traffic based DLinear (horizon 96), the metric is MSE.
>
> - For datasets with pronounced seasonality, such as traffic, a larger score weight *β* ($\beta$=0.8 or  0.9) (i.e., considering a higher proportion of the seasonal component in the loss calculation) yields better performance.
>
> - For datasets with pronounced trend, such as ETTh2, a small score weight *β* ($\beta$=0.1or 0.2) yields better performance.
>
>
> **W2&Q1:** The paper would be strengthened by a deeper discussion of such theoretical grounding.
>
> Thank you for your insightful feedback and constructive observations regarding our work. In order to better explain the differences between DBLoss and MSE. We provide a theoretical analysis explaining why DBLoss is more effective than MSE in this context. Motivated by the success of methods such as Dlinear [1], DUET [2], TimeMixer [3], and xPatch [4], which model time series by decomposing them into trend and seasonal components, achieving excellent performance, we assume that the trend and seasonal components are highly independent.
>
> We represent the time series as $y_t$, which can be decomposed into a trend component $T_t$ and a seasonal component $S_t$, such that:
>
> $y_t = T_t + S_t$
>
> For the model’s prediction $\hat{y}_t$, we similarly have:
>
> $\hat{y}_t = \hat{T}_t + \hat{S}_t$
>
> Under this setting, the Mean Squared Error (MSE) can be expanded as:
>
> $$L _{MSE} = ||y _t - \hat{y} _t|| _2 ^2 = ||(T _t + S _t) - (\hat{T} _t + \hat{S} _t)||_2 ^2$$
> $$  ~~~~~~~~~~ = ||y _t - \hat{y} _t|| _2^2 = ||(T _t - \hat{T}_t) + (S _t - \hat{S} _t)||_2^2$$
> $$ ~~~~~~~~~~ = ||T _t - \hat{T}_t|| _2^2 + ||S _t - \hat{S}_t||_2 ^2 + 2 \cdot (T_t - \hat{T}_t) \cdot (S_t - \hat{S}_t)$$
>
> The key part from MSE lies in the cross term $2 \cdot (T_t - \hat{T}_t) \cdot (S_t - \hat{S}_t)$. Our assumption is that the trend and seasonal components are highly independent.However, this cross term introduces an interaction between them, potentially making it difficult for the model to optimize the two components independently, which can degrade the overall prediction performance. For instance, if the trend component is poorly predicted while the seasonal component is well captured, the interaction term can still yield a large negative value of $2 \cdot (T_t - \hat{T}_t) \cdot (S_t - \hat{S}_t)$, disproportionately affecting the total loss.
>
> Furthermore, we analyze $L_{MSE}$ from the perspective of differentiation, and reveal MSE loss function being unable to independently consider these two components during the optimization process:
>
> According to the chain rule:
>
> $\nabla _{\boldsymbol{\Theta}} L _t = 2 \cdot [(T _t - \hat{T}_t) + (S _t - \hat{S}_t)] \cdot \nabla _{\boldsymbol{\Theta}} (-\hat{T}_t - \hat{S}_t) = -2 \cdot [(T _t - \hat{T}_t) + (S _t - \hat{S}_t)] \cdot \left( \nabla _{\boldsymbol{\Theta}} \hat{T}_t + \nabla _{\boldsymbol{\Theta}} \hat{S}_t \right)$
>
> Using Jacobian matrix notation:
>
> - Jacobian of Trend: $$ \mathbf{J} _T := \nabla _{\boldsymbol{\Theta}} \hat{T} _t  $$
> - Jacobian of Seasonality: $$ \mathbf{J}_S := \nabla _{\boldsymbol{\Theta}} \hat{S}_t $$
>
> Then:
>
> $$ \nabla_{\boldsymbol{\Theta}} L_t = -2 \cdot [(T_t - \hat{T}_t) + (S_t - \hat{S}_t)] \cdot (\mathbf{J}_T + \mathbf{J}_S) $$
>
> $$ \nabla_{\boldsymbol{\Theta}} L_t = -2 \cdot \left[ (T_t - \hat{T}_t)(\mathbf{J}_T + \mathbf{J}_S) + (S_t - \hat{S}_t)(\mathbf{J}_T + \mathbf{J}_S) \right] $$
>
> Expanding the terms:
>
> $$ \nabla_{\boldsymbol{\Theta}} L_t = -2 \cdot \left[ (T_t - \hat{T}_t)\mathbf{J}_T + (T_t - \hat{T}_t)\mathbf{J}_S + (S_t - \hat{S}_t)\mathbf{J}_T + (S_t - \hat{S}_t)\mathbf{J}_S \right] $$
>
> We split this into two parts:
>
> 1、Ideal Decoupled Term
>
> $$ \mathbf{G}_{\text{ideal}} = -2 \cdot \left[ (T_t - \hat{T}_t)\mathbf{J}_T + (S_t - \hat{S}_t)\mathbf{J}_S \right] $$
>
> 2、Coupled Term
>
> $$ \mathbf{G}_{\text{coupling}} = -2 \cdot \left[ (T_t - \hat{T}_t)\mathbf{J}_S + (S_t - \hat{S}_t)\mathbf{J}_T \right] $$
>
> This term represents the problematic coupling:
>
> - The trend error influences the seasonality‑guided optimization.
> - The seasonal error influences the trend‑guided optimization.
>
> The full gradient is expressed as the sum of the ideal and coupled terms: $$ \nabla _{\boldsymbol{\Theta}} L _t = \mathbf{G} _{\text{ideal}} + \mathbf{G} _{\text{coupling}} $$
>
> As long as $T _t \neq \hat{T}_t $ or $S _t  \neq \hat{S} _t $ (i.e., the model has not fully converged), the coupled term $  \mathbf{G} _{\text{coupling}}  \neq \mathbf{0} $. Due to the presence of $ \mathbf{G} _{\text{coupling}} $, the MSE loss function being unable to independently consider these two components during the optimization process.
>
>
>
> Compared with MSE, DBLoss ($L _{DB} = \lambda_1 ||T _t - \hat{T} _t||_1 + \lambda_2 ||S _t - \hat{S}_t||_2^2$ , where $\lambda_1$ and $\lambda_2$ are hyperparameters controlling the weights of the trend and seasonal losses, respectively.) removes this cross term (or $  \mathbf{G} _{\text{coupling}} $), enabling the model to treat these two components independently throughout the optimization process.   Furthermore, by adjusting the coefficients or using different distance norms, one can precisely control the loss scale for each component, enabling targeted learning and better modeling of both parts.
>
> [1] Are Transformers Effective for Time Series Forecasting? AAAI, 2023
>
> [2] DUET: Dual Clustering Enhanced Multivariate Time Series Forecasting, SIGKDD, 2025
>
> [3] TimeMixer: Decomposable Multiscale Mixing for Time Series Forecasting, ICLR, 2024
>
> [4] xPatch: Dual-Stream Time Series Forecasting with Exponential Seasonal-Trend Decomposition, AAAI, 2025
>
> **W3:** DBLoss does not explicitly account for residuals or noise components after decomposition.
>
> Thank you for your valuable feedback. We deeply appreciate your thoughtful observations. The reason we did not explicitly consider the residual term in our methodology is that we believe the residual is typically an unpredictable random component. Since residuals often contain noise and irregular patterns, we think that explicitly modeling or penalizing the residual during the decomposition process wouldn't be very meaningful and may even have negative consequences.
>
> Instead, we believe that the foundation of time series forecasting lies in the relative stability and predictability of the data. Therefore, we focus on the trend and seasonal components, which are the primary driving forces behind the predictable patterns in time series data. By using the Exponential Moving Average (EMA) for decomposition, we aim to effectively capture these components.

---

> > ### Author Response · Authors · 2025-08-01
> > **Results of Tables 1 using the modified DBLoss.**
> >
> > We tested Table 1 again using the modified code.
> >
> > | Model         |      | iTransformer |       |            |           | TimeMixer |       |            |           |
> > | ------------- | ---- | ------------ | ----- | ---------- | --------- | --------- | ----- | ---------- | --------- |
> > | Loss Function |      | Ori          |       | DBLoss-new |           | Ori       |       | DBLoss-new |           |
> > | Metric        |      | MSE          | MAE   | MSE        | MAE       | MSE       | MAE   | MSE        | MAE       |
> > | ETTh1         | Avg  | 0.439        | 0.448 | **0.423**  | **0.430** | 0.427     | 0.441 | **0.411**  | **0.429** |
> > | ETTh2         | Avg  | 0.370        | 0.403 | **0.363**  | **0.395** | 0.349     | 0.397 | **0.346**  | **0.387** |
> > | ETTm1         | Avg  | 0.361        | 0.390 | **0.350**  | **0.376** | 0.356     | 0.380 | **0.352**  | **0.375** |
> > | ETTm2         | Avg  | 0.269        | 0.327 | **0.261**  | **0.317** | 0.257     | 0.318 | **0.251**  | **0.307** |
> > | Solar         | Avg  | **0.202**    | 0.262 | 0.203      | **0.237** | **0.193** | 0.252 | **0.193**  | **0.229** |
> > | Weather       | Avg  | 0.232        | 0.270 | **0.229**  | **0.261** | 0.226     | 0.264 | **0.222**  | **0.254** |
> > | Electricity   | Avg  | 0.163        | 0.258 | **0.160**  | **0.252** | 0.185     | 0.284 | **0.184**  | **0.282** |
> > | Traffic       | Avg  | **0.397**    | 0.281 | **0.397**  | **0.278** | 0.409     | 0.279 | **0.400**  | **0.261** |
> >
> >
> >
> >
> >
> > | Model         |      | PatchTST |       |            |           | DLinear   |       |            |           |
> > | ------------- | ---- | -------- | ----- | ---------- | --------- | --------- | ----- | ---------- | --------- |
> > | Loss Function |      | Ori      |       | DBLoss-new |           | Ori       |       | DBLoss-new |           |
> > | Metric        |      | MSE      | MAE   | MSE        | MAE       | MSE       | MAE   | MSE        | MAE       |
> > | ETTh1         | Avg  | 0.419    | 0.436 | **0.402**  | **0.420** | 0.425     | 0.439 | **0.414**  | **0.428** |
> > | ETTh2         | Avg  | 0.351    | 0.395 | **0.339**  | **0.383** | 0.470     | 0.468 | **0.391**  | **0.414** |
> > | ETTm1         | Avg  | 0.349    | 0.381 | **0.344**  | **0.368** | 0.356     | 0.378 | **0.351**  | **0.371** |
> > | ETTm2         | Avg  | 0.256    | 0.314 | **0.252**  | **0.307** | 0.259     | 0.324 | **0.256**  | **0.313** |
> > | Solar         | Avg  | 0.200    | 0.284 | **0.187**  | **0.233** | **0.224** | 0.286 | 0.228      | **0.243** |
> > | Weather       | Avg  | 0.224    | 0.262 | **0.223**  | **0.252** | 0.242     | 0.293 | **0.239**  | **0.280** |
> > | Electricity   | Avg  | 0.171    | 0.270 | **0.170**  | **0.266** | 0.167     | 0.264 | **0.166**  | **0.261** |
> > | Traffic       | Avg  | 0.397    | 0.275 | **0.394**  | **0.266** | 0.418     | 0.287 | **0.417**  | **0.280** |

---

> > > ### Comment · Reviewer_PehU · 2025-08-05
> > >
> > > The authors have provided a comprehensive reply accompanied by additional experimental evidence, which effectively addresses the questions raised in the initial review. If these clarifications and supporting results are properly reflected in the revised manuscript, they are expected to improve the presentation and reinforce the empirical validity of the work.

---

> ### Author Response · Authors · 2025-08-05
> **Dear Reviewer PehU, Thank you for recognizing our work and valuable feedback.**
>
> Dear Reviewer PehU, thank you very much for your positive and encouraging feedback. We are delighted to hear that you found our reply comprehensive and the additional experimental evidence effective in addressing your initial questions.
> Following your advice, we have carefully integrated all the clarifications and supporting results into the revised manuscript. Thank you once again for your valuable time and guidance, which have been instrumental in improving our paper.

---

> ### Author Response · Authors · 2025-08-05
> **Dear Reviewer PehU,  could you consider raising the score?**
>
> Dear reviewer PehU, thank you again for your sincere and positive comments.
> We promise to incorporate all clarifications and supporting results into the revised draft. Could you consider raising the scores?

---

### Official Review · Reviewer_GNuv · 2025-06-29

**Clarity:** 3
**Significance:** 3
**Originality:** 3
**Rating:** 5
**Confidence:** 3

**Summary:**

This study proposes a simple yet effective Decomposition-Based Loss function called DBLoss. This method uses exponential moving averages to decompose the time series into seasonal and trend components within the forecasting horizon, and then calculates the loss for each of these components separately, followed by weighting them.

**Questions:**

no.

**Ethical Concerns:**

["NO or VERY MINOR ethics concerns only"]

**Final Justification:**

My concerns have been addressed. Regarding the additional theoretical analysis, I agree with the hypothesis and the corresponding theoretical insights.

**Limitations:**

YES

**Quality:**

3

**Strengths And Weaknesses:**

S1. Multivariate time series forecasting is important to time-series analysis.

S2. This work focuses on an important problem that could have real-world applications.

S3. The tables and figures used in this work are clear and easy to read.

S4. Experimental results demonstrate that DBLoss consistently outperforms state-of-the-art baselines across multiple real-world datasets.

---

W1. Why was EMA (Exponential Moving Average) decomposition chosen over other similar decomposition mechanisms? Providing corresponding explanations would make the paper more comprehensive.

W2. This infamous "ETT long term forecasting benchmark" is often criticized for its flaws such as limited domain coverage and the practice of forecasting at unreasonable horizons (e.g., 720 days into the future for exchange rate or oil temperature of a transformer at a specific hour months into the future). Every new model somehow beats this benchmark; however, there is still barely any absolute progress, only an illusion of it. Please refer to the talk (and paper) from Christoph Bergmeir [1, 2] where he discusses the limitation of this benchmark and current evaluation practices. A very recent position paper [3] also conducted a comprehensive evaluation of models on this benchmark showing that there's no obvious winner.

One (not so difficult) way to improve the quality of evaluation is to include results on better benchmarks.

---

Minor. Some citations are nonstandard by referencing arXiv preprints. Please ensure all citations point to formally published versions.

[1] https://neurips.cc/virtual/2024/workshop/84712#collapse108471

[2] Hewamalage, Hansika, Klaus Ackermann, and Christoph Bergmeir. "Forecast evaluation for data scientists: common pitfalls and best practices." Data Mining and Knowledge Discovery 37.2 (2023): 788-832.

[3] Brigato, Lorenzo, et al. "Position: There are no Champions in Long-Term Time Series Forecasting." arXiv preprint arXiv:2502.14045 (2025).

---

> ### Author Rebuttal · Authors · 2025-07-30
>
> Dear  **Reviewer GNuv**, thank you for providing your detailed and constructive feedback.
>
> **W1:** Why was EMA (Exponential Moving Average) decomposition chosen over other similar decomposition mechanisms? Providing corresponding explanations would make the paper more comprehensive.
>
> Thank you for your valuable feedback.  Our main contribution lies in proposing a new loss paradigm that, through a reasonable decomposition approach, enhances model performance. In fact, any decomposition method could be applied, but the quality of the decomposition directly affects the final performance. We adopted the currently advanced EMA decomposition method [1] with the aim of improving performance.
>
> [1] xPatch: Dual-Stream Time Series Forecasting with Exponential Seasonal-Trend Decomposition, AAAI, 2025
>
> **W2:** This infamous "ETT long term forecasting benchmark" is often criticized for its flaws such as limited domain coverage and the practice of forecasting at unreasonable horizons.Every new model somehow beats this benchmark; however, there is still barely any absolute progress, only an illusion of it.
>
> Thank you for your valuable feedback. To demonstrate the effectiveness of DBLoss, we conducted experiments on eight datasets from the widely recognized and authoritative TFB benchmark [2]. The results indicate that DBLoss significantly improves performance. We are reporting the average values for four prediction horizons for each dataset.
>
> [2] TFB: Towards Comprehensive and Fair Benchmarking of Time Series Forecasting Methods, PVLDB, 2024, Best Paper Award Nomination.
>
> |     Model     |      | iTransformer |       |            |           | TimeMixer |       |            |           | PatchTST  |       |            |           |  DLinear  |       |            |           |
> | :-----------: | :--: | :----------: | :---: | :--------: | :-------: | :-------: | :---: | :--------: | :-------: | :-------: | :---: | :--------: | :-------: | :-------: | :---: | :--------: | :-------: |
> | Loss Function |      |     Ori      |       |   DBLoss   |           |    Ori    |       |   DBLoss   |           |    Ori    |       |   DBLoss   |           |    Ori    |       |   DBLoss   |           |
> |    Metric     |      |     MSE      |  MAE  |    MSE     |    MAE    |    MSE    |  MAE  |    MSE     |    MAE    |    MSE    |  MAE  |    MSE     |    MAE    |    MSE    |  MAE  |    MSE     |    MAE    |
> |      ILI      | Avg  |    1.857     | 0.892 | **1.765**  | **0.838** |   1.820   | 0.886 | **1.651**  | **0.798** |   1.902   | 0.879 | **1.900**  | **0.878** |   2.185   | 1.040 | **2.004**  | **0.979** |
> |   AQShunyi    | Avg  |    0.706     | 0.506 | **0.701**  | **0.500** | **0.706** | 0.506 |   0.707    | **0.499** |   0.703   | 0.507 | **0.700**  | **0.502** |   0.706   | 0.522 | **0.704**  | **0.512** |
> |     AQWan     | Avg  |    0.809     | 0.496 | **0.804**  | **0.487** |   0.810   | 0.495 | **0.807**  | **0.486** |   0.812   | 0.498 | **0.810**  | **0.486** |   0.818   | 0.512 | **0.817**  | **0.502** |
> |    ZafNoo     | Avg  |    0.523     | 0.456 | **0.519**  | **0.453** |   0.518   | 0.451 | **0.515**  | **0.440** | **0.509** | 0.454 | **0.509**  | **0.441** |   0.496   | 0.452 | **0.492**  | **0.446** |
> |    CzeLan     | Avg  |  **0.218**   | 0.272 | **0.218**  | **0.259** |   0.218   | 0.268 | **0.208**  | **0.244** |   0.222   | 0.274 | **0.219**  | **0.249** |   0.285   | 0.343 | **0.252**  | **0.288** |
> |    Covid19    | Avg  |    1.488     | 0.050 | **1.431**  | **0.047** |  12.941   | 0.133 | **9.755**  | **0.083** |   1.607   | 0.056 | **1.528**  | **0.051** |   8.074   | 0.239 | **5.721**  | **0.135** |
> |    FRED-MD    | Avg  |    75.609    | 1.437 | **63.029** | **1.335** |  76.156   | 1.439 | **65.032** | **1.355** |  78.174   | 1.478 | **66.261** | **1.350** |  96.092   | 1.693 | **82.883** | **1.489** |
> |      NN5      | Avg  |    0.660     | 0.550 | **0.657**  | **0.549** |   0.656   | 0.556 | **0.654**  | **0.554** |   0.689   | 0.590 | **0.687**  | **0.589** | **0.692** | 0.580 |   0.693    | **0.578** |
>
> **W3:** Minor. Some citations are nonstandard by referencing arXiv preprints. Please ensure all citations point to formally published versions.
>
> Thank you for your feedback. We appreciate your attention to detail. We understand the importance of citing formally published versions, and we will make sure to replace all references to arXiv preprints with their corresponding formally published versions.
>
>
>
> **Additional theoretical analysis:**
>
> Considering that we had not previously provided a theoretical analysis of DBLoss, we have now included this additional analysis to make our paper more complete.
>
>  Motivated by the success of methods such as Dlinear [1], DUET [2], TimeMixer [3], and xPatch [4], which model time series by decomposing them into trend and seasonal components, achieving excellent performance, we assume that the trend and seasonal components are highly independent.
>
> We represent the time series as $y_t$, which can be decomposed into a trend component $T_t$ and a seasonal component $S_t$, such that:
>
> $y_t = T_t + S_t$
>
> For the model’s prediction $\hat{y}_t$, we similarly have:
>
> $\hat{y}_t = \hat{T}_t + \hat{S}_t$
>
> Under this setting, the Mean Squared Error (MSE) can be expanded as:
>
> $$L _{MSE} = ||y _t - \hat{y} _t|| _2 ^2 = ||(T _t + S _t) - (\hat{T} _t + \hat{S} _t)||_2 ^2$$
> $$  ~~~~~~~~~~ = ||y _t - \hat{y} _t|| _2^2 = ||(T _t - \hat{T}_t) + (S _t - \hat{S} _t)||_2^2$$
> $$ ~~~~~~~~~~ = ||T _t - \hat{T}_t|| _2^2 + ||S _t - \hat{S}_t||_2 ^2 + 2 \cdot (T_t - \hat{T}_t) \cdot (S_t - \hat{S}_t)$$
>
> The key part from MSE lies in the cross term $2 \cdot (T_t - \hat{T}_t) \cdot (S_t - \hat{S}_t)$. Our assumption is that the trend and seasonal components are highly independent.However, this cross term introduces an interaction between them, potentially making it difficult for the model to optimize the two components independently, which can degrade the overall prediction performance. For instance, if the trend component is poorly predicted while the seasonal component is well captured, the interaction term can still yield a large negative value of $2 \cdot (T_t - \hat{T}_t) \cdot (S_t - \hat{S}_t)$, disproportionately affecting the total loss.
>
> Furthermore, we analyze $L_{MSE}$ from the perspective of differentiation, and reveal MSE loss function being unable to independently consider these two components during the optimization process:
>
> According to the chain rule:
>
> $\nabla _{\boldsymbol{\Theta}} L _t = 2 \cdot [(T _t - \hat{T}_t) + (S _t - \hat{S}_t)] \cdot \nabla _{\boldsymbol{\Theta}} (-\hat{T}_t - \hat{S}_t) = -2 \cdot [(T _t - \hat{T}_t) + (S _t - \hat{S}_t)] \cdot \left( \nabla _{\boldsymbol{\Theta}} \hat{T}_t + \nabla _{\boldsymbol{\Theta}} \hat{S}_t \right)$
>
> Using Jacobian matrix notation:
>
> - Jacobian of Trend: $$ \mathbf{J} _T := \nabla _{\boldsymbol{\Theta}} \hat{T} _t  $$
> - Jacobian of Seasonality: $$ \mathbf{J}_S := \nabla _{\boldsymbol{\Theta}} \hat{S}_t $$
>
> Then:
>
> $$ \nabla_{\boldsymbol{\Theta}} L_t = -2 \cdot [(T_t - \hat{T}_t) + (S_t - \hat{S}_t)] \cdot (\mathbf{J}_T + \mathbf{J}_S) $$
>
> $$ \nabla_{\boldsymbol{\Theta}} L_t = -2 \cdot \left[ (T_t - \hat{T}_t)(\mathbf{J}_T + \mathbf{J}_S) + (S_t - \hat{S}_t)(\mathbf{J}_T + \mathbf{J}_S) \right] $$
>
> Expanding the terms:
>
> $$ \nabla_{\boldsymbol{\Theta}} L_t = -2 \cdot \left[ (T_t - \hat{T}_t)\mathbf{J}_T + (T_t - \hat{T}_t)\mathbf{J}_S + (S_t - \hat{S}_t)\mathbf{J}_T + (S_t - \hat{S}_t)\mathbf{J}_S \right] $$
>
> We split this into two parts:
>
> 1、Ideal Decoupled Term
>
> $$ \mathbf{G}_{\text{ideal}} = -2 \cdot \left[ (T_t - \hat{T}_t)\mathbf{J}_T + (S_t - \hat{S}_t)\mathbf{J}_S \right] $$
>
> 2、Coupled Term
>
> $$ \mathbf{G}_{\text{coupling}} = -2 \cdot \left[ (T_t - \hat{T}_t)\mathbf{J}_S + (S_t - \hat{S}_t)\mathbf{J}_T \right] $$
>
> This term represents the problematic coupling:
>
> - The trend error influences the seasonality‑guided optimization.
> - The seasonal error influences the trend‑guided optimization.
>
> The full gradient is expressed as the sum of the ideal and coupled terms: $$ \nabla _{\boldsymbol{\Theta}} L _t = \mathbf{G} _{\text{ideal}} + \mathbf{G} _{\text{coupling}} $$
>
> As long as $T _t \neq \hat{T}_t $ or $S _t  \neq \hat{S} _t $ (i.e., the model has not fully converged), the coupled term $  \mathbf{G} _{\text{coupling}}  \neq \mathbf{0} $. Due to the presence of $ \mathbf{G} _{\text{coupling}} $, the MSE loss function being unable to independently consider these two components during the optimization process.
>
>
>
> Compared with MSE, DBLoss ($L _{DB} = \lambda_1 ||T _t - \hat{T} _t||_1 + \lambda_2 ||S _t - \hat{S}_t||_2^2$ , where $\lambda_1$ and $\lambda_2$ are hyperparameters controlling the weights of the trend and seasonal losses, respectively.) removes this cross term (or $  \mathbf{G} _{\text{coupling}} $), enabling the model to treat these two components independently throughout the optimization process.   Furthermore, by adjusting the coefficients or using different distance norms, one can precisely control the loss scale for each component, enabling targeted learning and better modeling of both parts.
>
> [1] Are Transformers Effective for Time Series Forecasting? AAAI, 2023
>
> [2] DUET: Dual Clustering Enhanced Multivariate Time Series Forecasting, SIGKDD, 2025
>
> [3] TimeMixer: Decomposable Multiscale Mixing for Time Series Forecasting, ICLR, 2024
>
> [4] xPatch: Dual-Stream Time Series Forecasting with Exponential Seasonal-Trend Decomposition, AAAI, 2025

---

> ### Comment · Reviewer_GNuv · 2025-08-04
>
> Thank you for your detailed response and the additional experimental results. My concerns have been addressed. Regarding the additional theoretical analysis you provided, I agree with your hypothesis and the corresponding theoretical insights, and as a result, I have decided to increase my evaluation score.

---

### Official Review · Reviewer_oJJG · 2025-07-02

**Clarity:** 3
**Significance:** 3
**Originality:** 3
**Rating:** 5
**Confidence:** 5

**Summary:**

This study proposes a simple yet effective loss function for time series forecasting, called DBLoss, which can refine the characterization and representation of time series through decomposition within the forecasting horizon. By decomposing the future sequence into different trend and seasonal components, DBLoss enhances the loss function's ability to improve prediction performance for different items.

**Questions:**

Q1: The end-to-end model in the paper was comprehensively evaluated on 8 benchmark datasets, while the foundation model was only validated on 4 datasets. Could you please explain the rationale behind this inconsistency in experimental design?

Q2: In Algorithm 1, the range of the EMA smoothing coefficient α is defined as (0, 1), suggesting that 0 and 1 are excluded. However, Figure 4 appears to include these boundary values. Could you clarify this discrepancy?

**Ethical Concerns:**

["NO or VERY MINOR ethics concerns only"]

**Final Justification:**

I appreciate the author's response and have accordingly increased my score.

**Limitations:**

The paper includes a dedicated section addressing the inherent limitations of the proposed approach.

**Quality:**

3

**Strengths And Weaknesses:**

S1. This paper is well motivated as it faces the challenge of effectively capturing the seasonality or trend within the forecasting horizon.. The notations are clear, the experimental results show that the proposed method is effective.

S2. This paper points out that the traditional MSE Loss sometimes fails to accurately capture the seasonality or trend within the forecasting horizon and designs a new Loss Function accordingly, providing a new perspective.


W1. The end-to-end model in the paper was comprehensively evaluated on 8 benchmark datasets, while the foundation model was only validated on 4 datasets. Could you please explain the rationale behind this inconsistency in experimental design?

W2. The paper employs EMA (Exponential Moving Average) decomposition. Why was STL (Seasonal-Trend decomposition using Loess) not adopted to decompose the series into trend, seasonal, and residual components, with separate loss calculations and weighted aggregation?

W3. In Algorithm 1, the range of the EMA smoothing coefficient α is defined as (0, 1), suggesting that 0 and 1 are excluded. However, Figure 4 appears to include these boundary values. Could you clarify this discrepancy?

W4. The proposed method is intuitively feasible, but a more fundamental theoretical explanation would help enhance the significance of the study. The study relies too heavily on empirical experimental analysis and lacks in-depth experimental justification.

---

> ### Author Rebuttal · Authors · 2025-07-30
>
> Dear  **Reviewer oJJG**, thank you for providing your detailed and constructive feedback.
>
> **W1&Q1:** The end-to-end model in the paper was comprehensively evaluated on 8 benchmark datasets, while the foundation model was only validated on 4 datasets.
>
> Thank you for your valuable feedback. We recognize the importance of consistency in experimental design. To address this, we have added additional experiments and extended the evaluation of the foundation model on more datasets to provide a more comprehensive comparison with the end-to-end model. The results of the additional experiments are as follows, and it is evident that DBLoss performs well on these datasets. We are reporting the average values for four prediction horizons for each dataset.
>
> |    Model    |      | GPT4TS |       |           |           |   CALF    |       |           |           |  Moment   |       |           |           |    TTM    |       |           |           |
> | :---------: | :--: | :----: | :---: | :-------: | :-------: | :-------: | :---: | :-------: | :-------: | :-------: | :---: | :-------: | :-------: | :-------: | :---: | :-------: | :-------: |
> |             |      |  Ori   |       |  DBLoss   |           |    Ori    |       |  DBLoss   |           |    Ori    |       |  DBLoss   |           |    Ori    |       |  DBLoss   |           |
> |             |      |  MSE   |  MAE  |    MSE    |    MAE    |    MSE    |  MAE  |    MSE    |    MAE    |    MSE    |  MAE  |    MSE    |    MAE    |    MSE    |  MAE  |    MSE    |    MAE    |
> |    Solar    | Avg  | 0.262  | 0.334 | **0.254** | **0.278** | **0.229** | 0.297 |   0.232   | **0.291** |   0.243   | 0.277 | **0.241** | **0.274** |   0.218   | 0.269 | **0.215** | **0.257** |
> |   Weather   | Avg  | 0.252  | 0.293 | **0.249** | **0.285** |   0.237   | 0.275 | **0.234** | **0.268** |   0.230   | 0.269 | **0.229** | **0.257** |   0.225   | 0.260 | **0.224** | **0.254** |
> | Electricity | Avg  | 0.207  | 0.316 | **0.205** | **0.311** |   0.172   | 0.268 | **0.170** | **0.264** | **0.193** | 0.296 | **0.193** | **0.294** | **0.179** | 0.276 | **0.179** | **0.274** |
> |   Traffic   | Avg  | 0.433  | 0.309 | **0.429** | **0.299** |   0.435   | 0.316 | **0.434** | **0.313** |   0.434   | 0.304 | **0.430** | **0.300** |   0.484   | 0.341 | **0.478** | **0.335** |
>
> **W2:** Why was STL (Seasonal-Trend decomposition using Loess) not adopted to decompose the series into trend, seasonal, and residual components, with separate loss calculations and weighted aggregation?
>
> Thank you for your valuable feedback. While STL is indeed a popular for time series decomposition, we chose to use EMA decomposition for the following reasons:
>
> - EMA provides a smooth and efficient way to model the trend without the computational complexity of Loess, which we found to be particularly slow. For example, when using the DLinear algorithm to predict 96 steps on the ETTh1 dataset, it required over 11 hours to run, severely limiting the practical usability of STL.
> - Our main contribution lies in proposing a new loss paradigm that, through a reasonable decomposition approach, enhances model performance. In fact, any decomposition method could be applied, but the quality of the decomposition directly affects the final performance. We adopted the currently advanced EMA decomposition method [1] with the aim of improving performance.
>
> [1] xPatch: Dual-Stream Time Series Forecasting with Exponential Seasonal-Trend Decomposition, AAAI, 2025
>
>
>
> **W3&Q2:** In Algorithm 1, the range of the EMA smoothing coefficient α is defined as (0, 1), suggesting that 0 and 1 are excluded. However, Figure 4 appears to include these boundary values. Could you clarify this discrepancy?
>
> Thank you for your valuable feedback. The range of α in the algorithm's definition is constrained to (0, 1), excluding 0 and 1, primarily to avoid computational errors such as division by zero when α = 1, and also to prevent the model from degenerating into a non-typical or meaningless smoothing method when α = 0. However, in Figure 4, to account for extreme cases, we manually replaced the value of 0 with an approximate value close to 0 and the value of 1 with an approximate value close to 1. We apologize for not clarifying this in the explanation of Figure 4. We will revise this in the corrected version of the paper.
>
>
>
> **W4:** The proposed method is intuitively feasible, but a more fundamental theoretical explanation would help enhance the significance of the study. The study relies too heavily on empirical experimental analysis and lacks in-depth experimental justification.
>
> Thank you for your insightful feedback and constructive observations regarding our work. In order to better explain the differences between DBLoss and MSE. We provide a theoretical analysis explaining why DBLoss is more effective than MSE in this context. Motivated by the success of methods such as Dlinear [1], DUET [2], TimeMixer [3], and xPatch [4], which model time series by decomposing them into trend and seasonal components, achieving excellent performance, we assume that the trend and seasonal components are highly independent.
>
> We represent the time series as $y_t$, which can be decomposed into a trend component $T_t$ and a seasonal component $S_t$, such that:
>
> $y_t = T_t + S_t$
>
> For the model’s prediction $\hat{y}_t$, we similarly have:
>
> $\hat{y}_t = \hat{T}_t + \hat{S}_t$
>
> Under this setting, the Mean Squared Error (MSE) can be expanded as:
>
> $$L _{MSE} = ||y _t - \hat{y} _t|| _2 ^2 = ||(T _t + S _t) - (\hat{T} _t + \hat{S} _t)||_2 ^2$$
> $$  ~~~~~~~~~~ = ||y _t - \hat{y} _t|| _2^2 = ||(T _t - \hat{T}_t) + (S _t - \hat{S} _t)||_2^2$$
> $$ ~~~~~~~~~~ = ||T _t - \hat{T}_t|| _2^2 + ||S _t - \hat{S}_t||_2 ^2 + 2 \cdot (T_t - \hat{T}_t) \cdot (S_t - \hat{S}_t)$$
>
> The key part from MSE lies in the cross term $2 \cdot (T_t - \hat{T}_t) \cdot (S_t - \hat{S}_t)$. Our assumption is that the trend and seasonal components are highly independent.However, this cross term introduces an interaction between them, potentially making it difficult for the model to optimize the two components independently, which can degrade the overall prediction performance. For instance, if the trend component is poorly predicted while the seasonal component is well captured, the interaction term can still yield a large negative value of $2 \cdot (T_t - \hat{T}_t) \cdot (S_t - \hat{S}_t)$, disproportionately affecting the total loss.
>
> Furthermore, we analyze $L_{MSE}$ from the perspective of differentiation, and reveal MSE loss function being unable to independently consider these two components during the optimization process:
>
> According to the chain rule:
>
> $\nabla _{\boldsymbol{\Theta}} L _t = 2 \cdot [(T _t - \hat{T}_t) + (S _t - \hat{S}_t)] \cdot \nabla _{\boldsymbol{\Theta}} (-\hat{T}_t - \hat{S}_t) = -2 \cdot [(T _t - \hat{T}_t) + (S _t - \hat{S}_t)] \cdot \left( \nabla _{\boldsymbol{\Theta}} \hat{T}_t + \nabla _{\boldsymbol{\Theta}} \hat{S}_t \right)$
>
> Using Jacobian matrix notation:
>
> - Jacobian of Trend: $$ \mathbf{J} _T := \nabla _{\boldsymbol{\Theta}} \hat{T} _t  $$
> - Jacobian of Seasonality: $$ \mathbf{J}_S := \nabla _{\boldsymbol{\Theta}} \hat{S}_t $$
>
> Then:
>
> $$ \nabla_{\boldsymbol{\Theta}} L_t = -2 \cdot [(T_t - \hat{T}_t) + (S_t - \hat{S}_t)] \cdot (\mathbf{J}_T + \mathbf{J}_S) $$
>
> $$ \nabla_{\boldsymbol{\Theta}} L_t = -2 \cdot \left[ (T_t - \hat{T}_t)(\mathbf{J}_T + \mathbf{J}_S) + (S_t - \hat{S}_t)(\mathbf{J}_T + \mathbf{J}_S) \right] $$
>
> Expanding the terms:
>
> $$ \nabla_{\boldsymbol{\Theta}} L_t = -2 \cdot \left[ (T_t - \hat{T}_t)\mathbf{J}_T + (T_t - \hat{T}_t)\mathbf{J}_S + (S_t - \hat{S}_t)\mathbf{J}_T + (S_t - \hat{S}_t)\mathbf{J}_S \right] $$
>
> We split this into two parts:
>
> 1、Ideal Decoupled Term
>
> $$ \mathbf{G}_{\text{ideal}} = -2 \cdot \left[ (T_t - \hat{T}_t)\mathbf{J}_T + (S_t - \hat{S}_t)\mathbf{J}_S \right] $$
>
> 2、Coupled Term
>
> $$ \mathbf{G}_{\text{coupling}} = -2 \cdot \left[ (T_t - \hat{T}_t)\mathbf{J}_S + (S_t - \hat{S}_t)\mathbf{J}_T \right] $$
>
> This term represents the problematic coupling:
>
> - The trend error influences the seasonality‑guided optimization.
> - The seasonal error influences the trend‑guided optimization.
>
> The full gradient is expressed as the sum of the ideal and coupled terms: $$ \nabla _{\boldsymbol{\Theta}} L _t = \mathbf{G} _{\text{ideal}} + \mathbf{G} _{\text{coupling}} $$
>
> As long as $T _t \neq \hat{T}_t $ or $S _t  \neq \hat{S} _t $ (i.e., the model has not fully converged), the coupled term $  \mathbf{G} _{\text{coupling}}  \neq \mathbf{0} $. Due to the presence of $ \mathbf{G} _{\text{coupling}} $, the MSE loss function being unable to independently consider these two components during the optimization process.
>
>
>
> Compared with MSE, DBLoss ($L _{DB} = \lambda_1 ||T _t - \hat{T} _t||_1 + \lambda_2 ||S _t - \hat{S}_t||_2^2$ , where $\lambda_1$ and $\lambda_2$ are hyperparameters controlling the weights of the trend and seasonal losses, respectively.) removes this cross term (or $  \mathbf{G} _{\text{coupling}} $), enabling the model to treat these two components independently throughout the optimization process.   Furthermore, by adjusting the coefficients or using different distance norms, one can precisely control the loss scale for each component, enabling targeted learning and better modeling of both parts.
>
> [1] Are Transformers Effective for Time Series Forecasting? AAAI, 2023
>
> [2] DUET: Dual Clustering Enhanced Multivariate Time Series Forecasting, SIGKDD, 2025
>
> [3] TimeMixer: Decomposable Multiscale Mixing for Time Series Forecasting, ICLR, 2024
>
> [4] xPatch: Dual-Stream Time Series Forecasting with Exponential Seasonal-Trend Decomposition, AAAI, 2025

---

> ### Comment · Reviewer_oJJG · 2025-08-01
>
> Thank you for your detailed response. My issue has been partially resolved, so I will increase my score.

---

> > ### Author Response · Authors · 2025-08-03
> > **Thank you for recognizing our work and increasing your score.**
> >
> > Dear Reviewer oJJG, thank you very much for your valuable suggestions on our work. We sincerely appreciate your feedback and are grateful for your willingness to improve the score, best wishes!

---

### Note · Authors · 2025-08-11

Dear Area Chair and reviewers:

Thanks again for your hard work and efforts! We are extremely grateful to all reviewers for their participation in the discussion, and we also appreciate the assistance from the Area Chair. During the discussion phase, **Reviewers oJJG and GNuv promise to improve their scores, and Reviewers PehU and 4VN9 believe that our response has addressed their initial review comments well**. We are pleased to see that all the issues have been effectively addressed. **We commit to incorporating all the clarifications and supporting results into the revised manuscript**, as per the suggestions of Reviewers PehU and 4VN9, thereby enhancing the clarity of the explanations and providing stronger experimental support for the contributions.

Authors of “DBLoss: Decomposition-based Loss Function for Time Series Forecasting”

Best Regards!

---

### Decision · Program_Chairs · 2025-09-17

**Decision:**

Accept (poster)

**Comment:**

The paper introduces a new loss, DBLoss, for time-series forecasting, which aims to properly capture seasonality and trend in forecasts. The paper has gone through several discussions between the authors and reviewers, and eventually, the final scores of A, A, BA, BA have been received. The quality of the paper has been increased throughout the rebuttal phase, and I support accepting the paper. I ask the authors to ensure that all the changes and additions requested by the authors be carefully addressed in the final version of the paper.